# Classification-by-Components: Probabilistic Modeling of Reasoning over a Set of Components

**Sascha Saralajew**[1,*]  **Lars Holdijk**[1,*]  **Maike Rees**[1]  **Ebubekir Asan**[1]  **Thomas Villmann**[2,*]

[1]Dr. Ing. h.c. F. Porsche AG, Weissach, Germany,
`sascha.saralajew@porsche.de`
[2]University of Applied Sciences Mittweida, Mittweida, Germany,
`thomas.villmann@hs-mittweida.de`

## Abstract

Neural networks are state-of-the-art classification approaches but are generally difficult to interpret. This issue can be partly alleviated by constructing a precise decision process within the neural network. In this work, a network architecture, denoted as Classification-By-Components network (CBC), is proposed. It is restricted to follow an intuitive reasoning based decision process inspired by BIEDERMAN's recognition-by-components theory from cognitive psychology. The network is trained to learn and detect generic components that characterize objects. In parallel, a class-wise reasoning strategy based on these components is learned to solve the classification problem. In contrast to other work on reasoning, we propose three different types of reasoning: positive, negative, and indefinite. These three types together form a probability space to provide a probabilistic classifier. The decomposition of objects into generic components combined with the probabilistic reasoning provides by design a clear interpretation of the classification decision process. The evaluation of the approach on MNIST shows that CBCs are viable classifiers. Additionally, we demonstrate that the inherent interpretability offers a profound understanding of the classification behavior such that we can explain the success of an adversarial attack. The method's scalability is successfully tested using the IMAGENET dataset.

## 1   Introduction

Neural Networks (NNs) dominate the field of machine learning in terms of image classification accuracy. Due to their design, considered as black boxes, it is however hard to gain insights into their decision making process and to interpret why they sometimes behave unexpectedly. In general, the interpretability of NNs is under controversial discussion [1–4] and pushed researchers to new methods to improve the weaknesses [5–7]. This is also highlighted in the topic of robustness of NNs against adversarial examples [8]. Prototype-based classifiers like Learning Vector Quantizers [9, 10] are more interpretable and can provide insights into their classification processes. Unfortunately, they are still hindered by their low base accuracies.

The method proposed in this work aims to answer the question of interpretability by drawing inspirations from BIEDERMAN's theory recognition-by-components [11] from the field of cognitive psychology. Roughly speaking, BIEDERMAN's theory describes how humans recognize complex objects by assuming that objects can be decomposed into generic parts that operate as structural primitives, called *components*. Objects are then classified by matching the *extracted decomposition plan* with a *class Decomposition Plan* (DP) for each potential object class. Intuitively, the class DPs

---

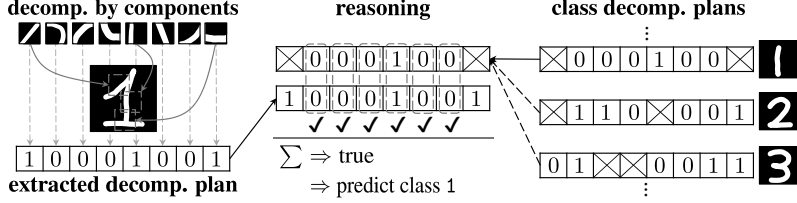

Figure 1: An example realization of the classification process of a CBC on a digit classification task. For simplicity, we illustrate a discrete case where "1" corresponds to detection / positive reasoning, "0" to no detection / negative reasoning, and "⊠" to indefinite reasoning.

describe which components are important to be detected and which components are important to not be detected for an object to belong to a specific class. For example, if we consider the classification of a digit as illustrated in Fig. 1, the detection of a component representing a vertical bar provides evidence in favor of the class 1. In other words, we *reason positively* over the vertical bar component for the class 1. Similarly, we can *reason negatively* over all curved components. In contrast to other work on reasoning, the presented approach extends these two intuitive reasoning states by a third type considering *indefinite reasoning*. In Fig. 1, not all components will be important for the recognition of a 1. For instance, we reason neither positively nor negatively over the serif and bottom stroke because not all writing styles use them. In Sec. 2, a network architecture is introduced that models the described classification process in an end-to-end trainable framework such that the components as well as the class DPs can be learned. In line with BIEDERMAN's theory, we call this a Classification-By-Components network (CBC).

In summary, the contribution of this paper is a classification method, called CBC, with the following important characteristics: **(1)** The method classifies its input by applying *positive, negative*, and *indefinite reasoning* over an extracted DP. To the best of our knowledge, this is the first time that optionality of components / features is explicitly modeled. **(2)** The method uses a probabilistic reasoning process that directly outputs class hypothesis probabilities without requiring heuristic squashing methods such as softmax. **(3)** The reasoning process is easily interpretable and simplifies the understanding of the classification decision. **(4)** The method retains advantages of NNs such as being end-to-end trainable on large scale datasets and achieving high accuracies on complex tasks.

## 2 The classification-by-components network

In the following, we will describe the CBC architecture and how to train it. We present the architecture using full-size components and consecutively generalize this to patch components. Both principles are used in the evaluation in Sec. 4. The architectures are defined (without loss of generality) for vectorial inputs but can be extended to higher dimensional inputs like images.

### 2.1 Reasoning over a set of full-size components

The proposed framework relies on a probabilistic model based on a probability tree diagram $T$. This tree $T$ can be decomposed into sub-trees $T_c$ for each class $c$ with the prior class probability $P(c)$ on the starting edge. Such a sub-tree is depicted in Fig. 2. The whole probability tree diagram is modeled over five random variables: $c$, indicator variable of the class; $k$, indicator variable of the component; $I$, binary random variable for importance; $R$, binary random variable for reasoning by detection; $D$, binary random variable for detection. The probabilities in the tree $T_c$ are interpreted in the following way: $P(k)$, probability that the $k$-th component occurs; $P(I|k,c)$, probability that the $k$-th component is important for the class $c$; $P(R|k,c)$, probability that the $k$-th component has to be detected for the class $c$; $P(D|k,\mathbf{x})$, probability that the $k$-th component is detected in the input $\mathbf{x}$. The horizontal bar indicates the complementary event, i.e. $P(\overline{D}|k,\mathbf{x})$ is the probability that the $k$-th component is *not* detected in the input $\mathbf{x}$. Based on these definitions we derive the CBC architecture.

**Extracting the decomposition plan** Given an input $\mathbf{x} \in \mathbb{R}^{n_x}$ and a set of trainable *full-size components* $\mathcal{K} = \{\boldsymbol{\kappa}_k \in \mathbb{R}^{n_\kappa} | k = 1, ..., \#\mathcal{K}\}$ with $n_x = n_\kappa$, the first part of the network detects

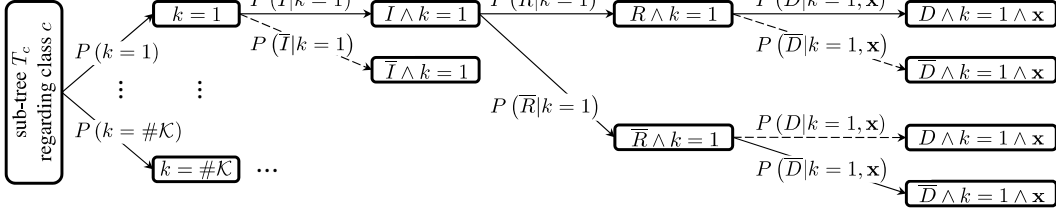

Figure 2: The probability tree diagram $T_c$ that represents the reasoning about a class $c$. For better readability, the variable of class $c$ is dropped in the mathematical expressions and we only show the full sub-tree for the first component. The solid line paths are the paths of agreement.

the presence of a component $\boldsymbol{\kappa}_k$ in $\mathbf{x}$. A feature extractor $\mathbf{f}(\mathbf{x}) = \mathbf{f}(\mathbf{x}; \boldsymbol{\theta})$ with trainable weights $\boldsymbol{\theta}$ takes an input and outputs a feature vector $\mathbf{f}(\mathbf{x}) \in \mathbb{R}^{m_x}$. The feature extractor is used in a Siamese architecture [12] to extract the features of the input $\mathbf{x}$ and of all the components $\{\mathbf{f}(\boldsymbol{\kappa}_k)\}_k$. The extracted features are used to measure the probability $P(D|k, \mathbf{x})$ for the detection of a component by a *detection probability function* $d_k(\mathbf{x}) = d(\mathbf{f}(\mathbf{x}), \mathbf{f}(\boldsymbol{\kappa}_k)) \in [0, 1]$ with the requirement that $\mathbf{f}(\mathbf{x}) = \mathbf{f}(\boldsymbol{\kappa}_k)$ implies $d_k(\mathbf{x}) = 1$. Examples of suitable detection probability functions are the negative exponential over the squared Euclidean distance or the cosine similarity with a suitable handling of its negative part. To finalize the first part of the network, the detection probabilities are collected into the extracted DP as a vector $\mathbf{d}(\mathbf{x}) = (d_1(\mathbf{x}), ..., d_{\#\mathcal{K}}(\mathbf{x}))^{\mathrm{T}} \in [0, 1]^{\#\mathcal{K}}$.

**Modeling of the class decomposition plans**    The second part of the network models the class DPs for each class $c \in \mathcal{C} = \{1, ..., \#\mathcal{C}\}$ using the three forms of reasoning discussed earlier. Therefore, we define the *reasoning probabilities*, $r_{c,k}^+$, $r_{c,k}^-$, and $r_{c,k}^0$ as trainable parameters of the model. *Positive reasoning* $r_{c,k}^+ = P(I, R|k, c)$: The probability that the $k$-th component is important and must be detected to support the class hypothesis $c$. *Negative reasoning* $r_{c,k}^- = P(I, \overline{R}|k, c)$: The probability that the $k$-th component is important and must *not* be detected to support the class hypothesis $c$. *Indefinite reasoning* $r_{c,k}^0 = P(\overline{I}|k, c)$: The probability that the $k$-th component is not important for the class hypothesis $c$.[2] Together they form a probability space and hence $r_{c,k}^+ + r_{c,k}^0 + r_{c,k}^- = 1$. All reasoning probabilities are collected class-wise into vectors $\mathbf{r}_c^+ = (r_{c,1}^+, ..., r_{c,\#\mathcal{K}}^+)^{\mathrm{T}} \in [0, 1]^{\#\mathcal{K}}$ and $\mathbf{r}_c^-$, $\mathbf{r}_c^0$, respectively.

**Reasoning**    We compute the class hypothesis probability $p_c(\mathbf{x})$ regarding the paths of agreement under the condition of importance. An agreement $A$ is a path in the tree $T$ where either a component is detected ($D$) and requires reasoning by detection ($R$), or a component is not detected ($\overline{D}$) and requires reasoning by no detection ($\overline{R}$). The paths of agreement are marked with solid lines in Fig. 2. Hence, we model $p_c(\mathbf{x})$ by $P(A|I, \mathbf{x}, c)$:

$$P(A|I, \mathbf{x}, c) = \frac{\sum_k \left( P(R|k, c) P(D|k, \mathbf{x}) + P(\overline{R}|k, c) P(\overline{D}|k, \mathbf{x}) \right) P(I|k, c) P(k)}{\sum_k \left( 1 - P(\overline{I}|k, c) \right) P(k)}.$$

Substituting by the short form notations for the probabilities, assuming that $P(k) = \frac{1}{\#\mathcal{K}}$, and rewriting it with matrix calculus yields

$$p_c(\mathbf{x}) = \frac{(\mathbf{d}(\mathbf{x}))^{\mathrm{T}} \cdot \mathbf{r}_c^+ + (\mathbf{1} - \mathbf{d}(\mathbf{x}))^{\mathrm{T}} \cdot \mathbf{r}_c^-}{\mathbf{1}^{\mathrm{T}} \cdot (\mathbf{1} - \mathbf{r}_c^0)} = (\mathbf{d}(\mathbf{x}))^{\mathrm{T}} \cdot \bar{\mathbf{r}}_c^+ + (\mathbf{1} - \mathbf{d}(\mathbf{x}))^{\mathrm{T}} \cdot \bar{\mathbf{r}}_c^-, \qquad (1)$$

where $\mathbf{1}$ is the one vector of dimension $\#\mathcal{K}$ and $\bar{\mathbf{r}}_c^{\pm}$ are the normalized *effective reasoning possibility vectors*. The probabilities for all classes are then collected into the *class hypothesis possibility vector* $\mathbf{p}(\mathbf{x}) = (p_1(\mathbf{x}), ..., p_{\#\mathcal{C}}(\mathbf{x}))^{\mathrm{T}}$ to create the network output. We emphasize that $\mathbf{p}(\mathbf{x})$ is a *possibility* vector as $\sum_c p_c(\mathbf{x}) = 1$ does not necessarily hold. See the supplementary material Sec. B.1 for a detailed derivation of Eq. (1) and Sec. B.2 for a transformation of $\mathbf{p}(\mathbf{x})$ into a class probability vector.

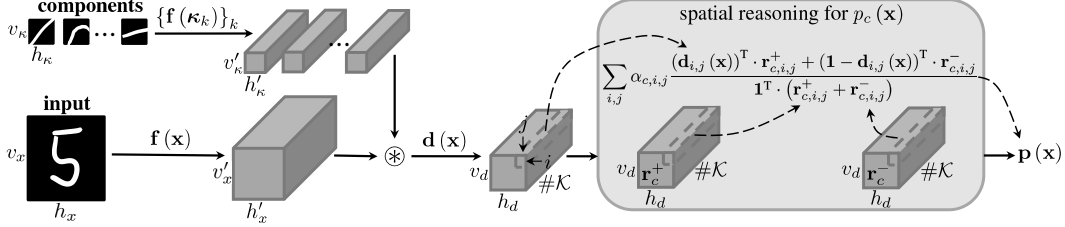

Figure 3: CBC with patch components and spatial reasoning for image inputs.

**Training of a CBC**  We train the networks end-to-end by minimizing the contrastive loss

$$l(\mathbf{x}, y) = \phi \left( \max \{ p_c(\mathbf{x}) \,|\, c \neq y, c \in \mathcal{C} \} - p_y(\mathbf{x}) \right) \tag{2}$$

where $y \in \mathcal{C}$ is the class label of $\mathbf{x}$, using stochastic gradient descent learning. The function $\phi$ : $[-1, 1] \rightarrow \mathbb{R}$ is a monotonically increasing, almost everywhere differentiable squashing function. It regulates the generalization-robustness-trade-off over the probability gap between the correct and highest probable incorrect class. This loss is similar to commonly used functions in prototype-based learning [14, 15]. The trainable parameters of a CBC are $\boldsymbol{\theta}$, all $\boldsymbol{\kappa} \in \mathcal{K}$, and $\mathbf{r}_c^+$, $\mathbf{r}_c^0$, $\mathbf{r}_c^-$ for all $c \in \mathcal{C}$. We refer to the supplementary material Sec. D for detailed information about the training procedure.

## 2.2   Extension to patch components

Assume the feature extractor $\mathbf{f}$ processes different input sizes down to a minimum (receptive field) dimension of $n_0$, similar to most Convolutional NNs (CNNs). To relax the assumption $n_x = n_\kappa$ of full-size components and to step closer to the motivating example of Fig. 1, we use a set $\mathcal{K}$ of trainable *patch components* with $n_x \geq n_\kappa \geq n_0$ such that $\mathbf{f}(\boldsymbol{\kappa}_k) \in \mathbb{R}^{m_\kappa}$ where $m_x \geq m_\kappa$. Moreover, $d_k(\mathbf{x})$ is extended to a sliding operation [16, 17], denoted as $\circledast$. The result is a *detection possibility stack (*extracted spatial DP) of size $v_d \times \#\mathcal{K}$ where $v_d$ is the spatial dimension after the sliding operation, see Fig. 3 for an image processing CBC. However, Eq. (1) can only handle one detection probability for each component and thus the reasoning process has to be redefined:

**Downsampling**  A simple approach is to downsample the detection possibility stack over the spatial dimension $v_d$ such that the output is a detection possibility vector and Eq. (1) can be applied. This can be achieved by applying global pooling techniques like global max pooling.

**Spatial reasoning**  Another approach is the extension of the reasoning process to work on the spatial DP which we call *spatial reasoning*. For this, the detection possibility stack of size $v_d \times \#\mathcal{K}$ is kept as depicted in Fig. 3. To compute the class hypothesis probabilities $p_c(\mathbf{x})$, Eq. (1) is redefined to be a weighted mean over the reasoning at each spatial position $i = 1, ..., v_d$. Thereby, $\alpha_{c,i} \in [0, 1]$ with $\sum_i \alpha_{c,i} = 1$ are the (non-)trainable *class-wise pixel probabilities* resembling the importance of each pixel position $i$. See the supplementary material in Sec. C for a further extension.

## 3   Related Work

**Reasoning in neural networks**  In its simplest form, one can argue that a NN already yields decisions based on reasoning. If one considers a NN to be entirely similar to a multilayer perceptron, the sign of each weight can be interpreted as either negative or positive reasoning over the corresponding feature. In this case, a weight of zero would model indefinite reasoning. However, the use of the Rectified Linear Unit (ReLU) activations forces NNs to be positive reasoning driven only. Nevertheless, this interpretation of the weights is used in interpretation techniques such as Class-Activation-Mapping (CAM) [5], which is similar to heatmap visualizations of CBCs.

**Explicit modeling of reasoning**  The use of components, and the inclusion of the negative and indefinite reasoning can be seen as an extension of the work in [7]. However, CBCs do not rely on the complicated three step training procedure presented in the paper and are built upon a probabilistic reasoning model. In [18], a form of reasoning is introduced similar to the indefinite reasoning state by occluding parts of the learned representation. Their components are, however, modeled in a textual form. In general, the reasoning process has slight similarities to ideas mentioned in [19] and the modeling of knowledge via graph structures [20–22].

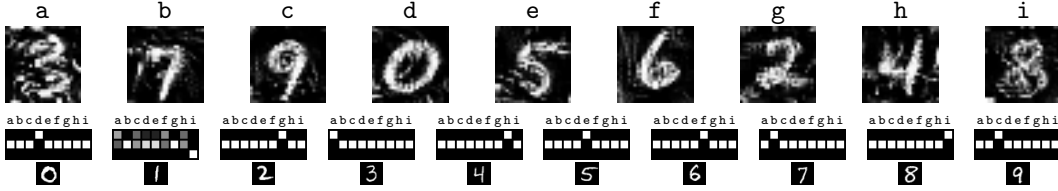

Figure 4: Learned reasoning process of a CBC with 9 components on MNIST. *Top row:* The learned components. *Bottom row:* The learned reasoning probabilities collected in reasoning matrices. The class is indicated by the MNIST digit below. The top row corresponds to $r_{c,k}^+$, middle row to $r_{c,k}^0$, and bottom row to $r_{c,k}^-$. White squares depict a probability of one and black squares of zero.

**Feature visualization**   If the components are defined as trainable parameters in the input space, then the learned components become similar to feature visualization techniques of NNs [23–25]. In contrast, the components are the *direct visualizations of the penultimate layer weights* (detection probability layer), are *not* computed via a post-processing, and have a probabilistic interpretation. Moreover, we are *not* applying regularizations to the components to resemble realistic images.

**Prototype-based classification rules and similarity learning**   A key ingredient of the proposed network is a Siamese architecture to learn a similarity measure [12, 26–28] and the idea to incorporate a kind of prototype-based classification rule into NNs [29–35]. Currently, the prototype[3] classification principle is gaining a lot of attention in few-shot learning due to its ability to learn fast from few data [29, 30, 36–38]. The idea to replace prototypes with patches in similarity learning has also been gaining attraction, as can be seen in [39] for the use of object tracking.

# 4   Evaluation

In this section, the evaluation of the CBCs is presented. Throughout the evaluation, interpretability is considered as an important characteristic. In this case, something is interpretable if it has a meaning to experts. We evaluate CBCs on MNIST [40] and IMAGENET [41]. The input spaces are defined over $[0, 1]$ and the datasets are normalized appropriately. Moreover, components that are defined in the input space are constrained to this space as well. The CBCs use the cosine similarity with ReLU activation as detection probability function. They are trained with the *margin loss* defined as Eq. (2) with $\phi(x) = \mathrm{ReLU}(x + \beta)$, where $\beta$ is a margin parameter, using the Adam optimizer [42]. An extended evaluation including an ablation study regarding the network setting on MNIST is presented in the supplementary material in Sec. E. Where possible, we report mean and standard deviation of the results. The source code is available at `www.github.com/saralajew/cbc_networks`.

## 4.1   MNIST

The CNN feature extractors are implemented without the use of batch normalization [43], with Swish activation [44], and the convolutional filters constraint to a Euclidean norm of one. We trained the components and reasoning probabilities from scratch using random initialization. Moreover, the margin parameter $\beta$ was set to $0.3$.

### 4.1.1   Negative reasoning: Beyond the best matching prototype principle

The CBC architecture in this experiment uses a 4-layer CNN feature extractor and full-size components. During the ablation study we found that in nearly all cases this CBC with 10 components converged to the Best Matching Prototype Principle (BMPP) [45] and formed prototypical components. This means that the reasoning for one class is performed with only strong positive reasoning over one and indefinite reasoning over all the other components, e. g. see the reasoning matrix of class 0 in Fig. 4 and the corresponding prototypical component d. To analyze if the network is able to classify using negative reasoning, we restricted the number of components to be smaller than the number of classes.

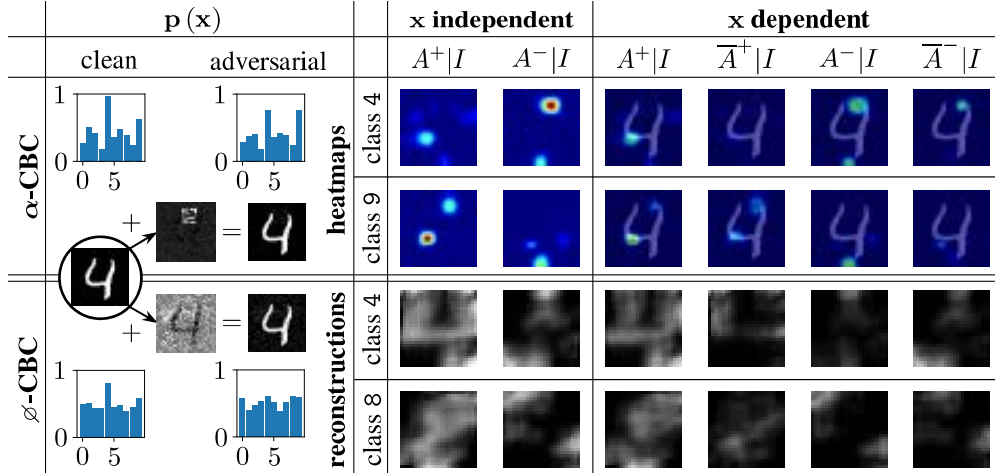

Figure 5: Visualization of the $\alpha$-CBC heatmaps and the $\varnothing$-CBC reconstructions for an adversarial input. For simplicity, we illustrate the more meaningful visualization for each model. The model visualizations correspond to the best matching reasoning stack regarding the input. We use the color coding "JET" to map probabilities of 0 to blue and 1 to red.

Fig. 4 shows the learned reasoning process of a CBC with 9 components. Similar to the 10 component version, the CBC learns to classify as many classes as possible by the BMPP. In the example, these are all classes except the class 1, for which the CBC uses weak positive reasoning over the components a, c, f, and h but mostly depends on negative reasoning over component i. This indicates that if an input image is classified as a 1, the network requires it to *not* look like an 8. A comparison of the shapes of the digits 1 and 8 supports this observation, the 8 only consists of curved edges while the 1 does not contain any and on average contains the least white pixels while the 8 requires the most. This result shows that by incorporating the negative and indefinite reasoning state, the CBCs are able to learn both the well understood BMPP and unrestricted approaches beyond the intuitive classification principles by themselves. Both networks achieved close to the state-of-the-art test accuracies over three runs of $(99.32 \pm 0.09)\%$.

### 4.1.2 Interpretation of the reasoning

In this section, we show the interpretability of CBCs. Similar to interpretation techniques from NNs we do this by considering input dependent and input independent visualizations. Moreover, to stress the visualizations in such a way that they really show how the model classifies, we: **(1)** Train two patch component CBCs similar to Fig. 3, one with trainable, denoted as $\alpha$-CBC, and one with non-trainable pixel probabilities fixed to $\alpha_{c,i,j} = (v_d \cdot h_d)^{-1}$, denoted as $\varnothing$-CBC. **(2)** Generate an adversarial image for both models with the boundary attack [46] and show how they fool the model.

Both CBCs use 8 patch components[4] of size $v_\kappa, h_\kappa = 7$. The feature extractor is a 2-layer CNN which extracts feature stacks of spatial size $v'_\kappa, h'_k = 1$ and $v'_x, h'_x = 22$. The spatial reasoning size of $v_d, h_d = 7$ was obtained by including a final max pooling operation of pool size 3 in $d(x)$. Additionally for each class, two reasoning possibility stacks were learned and winner-take-all was applied to determine $p_c(x)$. We call this *multiple reasoning* as we allow the model to learn multiple concepts for each class. The final test accuracies of both models are quasi equivalent and on average over three runs $(97.33 \pm 0.19)\%$. Similar to the previous section, the patch components start to resemble realistic digit parts like strokes, arcs, line-endings, etc.

The interpretability of the CBCs is based on visualizations of how the probability mass is distributed over the tree $T$. The class hypothesis probability $p_c(x)$, see Eq. (1), is the probability of *agreement under the condition of importance,* denoted by $A|I$. This event describes the correct matching of the extracted and class DP. Moreover, we decompose this event into the positive and negative reasoning part: *Positive $A|I$* is the event that a component is detected that should be detected and is denoted by

$A^+|I$. *Negative $A|I$* is the event that a component that should not be detected is not detected and is denoted by $A^-|I$. Both events can be related to paths in the trees $T_c$ from the root to the leaves, i.e. $A^+|I$ is the upper solid line path and $A^-|I$ is the lower solid line path in Fig. 2. The probability of $A|I$ can be thought of as evidence in favor of a class. Similarly, we can consider the complementary event of $A|I$ which is *disagreement under the condition of importance,* denoted by $\overline{A}|I$, and occurs when the extracted DP does not match the class DP. Again, this occurs either as *positive $\overline{A}|I$* when a component over which the CBC reasons positively is not detected, denoted by $\overline{A}^+|I$, or as *negative $\overline{A}|I$* when a component with negative reasoning is detected, denoted by $\overline{A}^-|I$. The related paths in the tree $T_c$ in Fig. 2 are the dashed line paths excluding non-importance. In general, the probability of $\overline{A}|I$ is evidence against a class.

Accordingly to Eq. (1), the visualizations are based on the probabilities in the tree $T$ for respective detection possibility vectors $\mathbf{z}_{i,j}$. These probabilities are collected into the following possibility vectors:[5] $\mathbf{z}_{i,j} \circ \bar{\mathbf{r}}^+_{c,i,j}$ for $A^+|I$; $(\mathbf{1} - \mathbf{z}_{i,j}) \circ \bar{\mathbf{r}}^+_{c,i,j}$ for $\overline{A}^+|I$; $(\mathbf{1} - \mathbf{z}_{i,j}) \circ \bar{\mathbf{r}}^-_{c,i,j}$ for $A^-|I$; $\mathbf{z}_{i,j} \circ \bar{\mathbf{r}}^-_{c,i,j}$ for $\overline{A}^-|I$. Moreover, we collect all the possibility vectors of one event for all $i, j$ in a stack. Using such a stack we create the visualizations by three procedures: **Probability heatmaps:** Upsample a stack to the input size and sum over $k$. This visualizes the probabilities for the respective event at the certain position. **Reconstructions:** Upsample a stack to $v'_x \times h'_x \times \#\mathcal{K}$, scale each patch component $\boldsymbol{\kappa}_k$ by the respective probability and draw them onto an initially black image of size $v_x \times h_x$ at the respective position. After a normalization step, the resulted reconstruction image gives an impression of the combination of the patches that is used to classify the image. **Incorporation of pixel probabilities:** Upsample the class-wise pixel probability maps $\alpha_c$ to $v_x \times h_x$ and normalize by the maximum value such that the most important pixels have a value of one. This map is finally overlaid over the heatmaps and reconstructions to highlight the impact of each pixel to the overall classification decision.

**Input independent interpretation** Input independent interpretations are calculated by setting $\mathbf{z}_{i,j}$ to the optimal vector with $\mathbf{1}$ for positive and for $\mathbf{0}$ negative $A|I$. They provide an answer to the question: "What has the model learned about the dataset?", see Fig. 5 "x independent". For both models, the learned concepts of the clean and adversarial class are visualized by the optimal $A^+|I$ and $A^-|I$. As visible in the heatmaps, the $\alpha$-CBC learned to recognize only as few parts as needed to distinguish the two classes. In case of the 4, this consists of a check that there is no stroke at the bottom and top, see $A^-|I$, while there is a corner on the left, see $A^+|I$. Such a radical sparse coding is learned for all classes. The reasoning for the 9 is similar except that it requires $A^+|I$ instead of $A^-|I$ for the top stroke. In contrast, the $\varnothing$-CBC learned the whole concept for digits and not just a sparse coding as the reconstructions show real digit shapes in the $A^+|I$. Moreover, the model performs interpretable "sanity checks" via $A^-|I$, e.g. no top stroke at the 4.

**Input dependent interpretation** Input dependent interpretations are obtained by setting $\mathbf{z}_{i,j}$ to $\mathbf{d}_{i,j}(\mathbf{x})$. To understand why the adversarial images fool the models by human imperceptible "noise" we answer the following question: "Which parts of the input provide evidence for/against the current classification decision?", see Fig. 5 "x dependent". By considering the clean probability histogram $\mathbf{p}(\mathbf{x})$ of the $\alpha$-CBC we see that the clean input perfectly fits the learned concept of a 4 as it had a probability of 1. The adversarial attack has turned the input into a 4 and 9 at the same time, see adversarial $\mathbf{p}(\mathbf{x})$. Remarkably, the attack found the high similarity between the two learned concepts and attacks the model by highlighting a few pixels in the top bar region in form of a patch – the manipulation only changes one pixel in $\mathbf{d}(\mathbf{x})$. Hence, the concept of a 4 is slightly violated as we see a highlighting of the top stroke region in the $\overline{A}^-|I$. This causes the probability drop of the class 4. At the same time, these few pixels provide $A^+|I$ for the top stroke of a 9 and, hence, raise the probability. For the $\varnothing$-CBC, the attack behavior is totally different. Since the clean input already does not match the learned concept perfectly as $p_4(\mathbf{x}) \approx 0.8$, the attack fools the model by reducing the contrast via background noise. For example, via the $\overline{A}^+|I$ the model highlights that the clear detection of the upper part of the 4 is not given. Moreover, it recognizes that there could be a top/bottom stroke, see $\overline{A}^-|I$. A similar interpretation holds for the adversarial class.

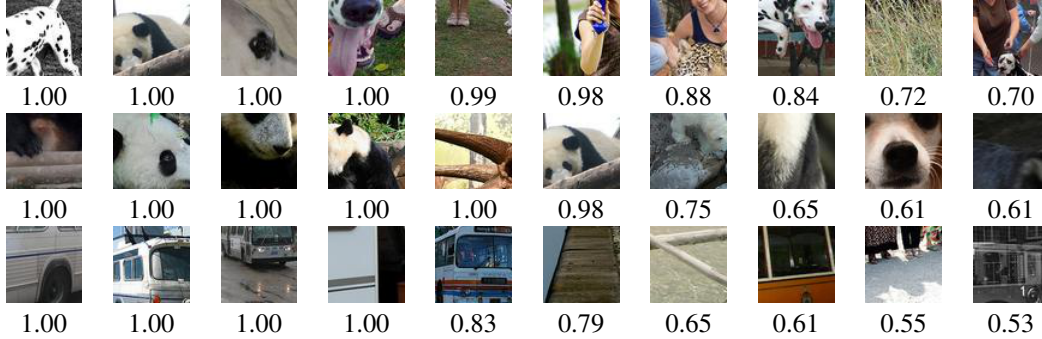

| 1.00 | 1.00 | 1.00 | 1.00 | 0.99 | 0.98 | 0.88 | 0.84 | 0.72 | 0.70 |
| 1.00 | 1.00 | 1.00 | 1.00 | 1.00 | 0.98 | 0.75 | 0.65 | 0.61 | 0.61 |
| 1.00 | 1.00 | 1.00 | 1.00 | 0.83 | 0.79 | 0.65 | 0.61 | 0.55 | 0.53 |

Figure 6: The 10 components with the highest $r_{c,k}^+$ for three different classes in the IMAGENET dataset. From top to bottom the classes are: dalmatian, giant panda, and trolleybus. Below each component the $r_{c,k}^+$ (rounded to two digits) is given with respect to the class in question.

**Overall result** The $\varnothing$-CBC with $\alpha_{c,i,j} = (v_d \cdot h_d)^{-1}$ is trained to learn a strong concept as it can only reach $p_y(\mathbf{x}) \approx 1$ if it reasons perfectly at *each* pixel position. Therefore, the probability histogram shows a relatively high base probability for all classes, as the overlap between encoded digits to a spatial size of $v_d, h_d = 7$ is often around 50%. Moreover, this restrictive classification principle violates the motivating example in Fig. 1 as the model cannot apply indefinite reasoning over a pixel region. In contrast, the $\alpha$-CBC is capable of modeling the motivating example but is at the same time a clear example of what happens if we optimize without any constraints as usually performed in NNs. Since the model is trained by minimizing an energy function, it learns to classify correctly with the lowest effort and, hence, oversimplifies. Therefore, the classification will be performed in a non-intuitive way. Moreover, the interpretation shows that the classification of both CBCs is based on non-robust features of $\mathbf{f}$ as both are highly sensitive to background manipulations.

## 4.2 IMAGENET

To evaluate CBCs on more complex data, we trained a CBC on the IMAGENET dataset. The CBC trained on IMAGENET was implemented using a pre-trained ResNet-50 [47] as *non-trainable* feature extractor. In contrast to the CBCs discussed earlier, the patch components of shape $m_\kappa = 2 \times 2 \times 2048$ are defined *directly* in the feature space. This removes the relation between the components and the input space but drastically improves training time. After downsampling the detection possibility stack of size $v_d, h_d = 6$ by global max pooling, the reasoning is applied, see Sec. 2.2. The components were initialized by cropping the center of 5 images from each class and consecutively processing them through the feature extractor, resulting in $5\,000$ patch components. If the component $\kappa_k$ was initialized by a sample from the class $c$, then we initialized $r_{c,k}^+$ as a uniform random value of $[z, 1]$ where $z = 0.75$ and as a uniform random value of $[0, 1-z]$ otherwise. Afterwards, the initialization of $r_{c,k}^-$ was determined by $r_{c,k}^+ \cdot \left(1 - r_{c,k}^+\right)$. Hence, we biased the model with positive reasoning to components that were sampled from the respective class. The CBC was trained with the margin loss and $\beta = 0.1$. In compliance with earlier work on IMAGENET, the input images were rescaled, by first rescaling the shortest side to 224 and then performing a center cropping of size $224 \times 224$. For the same reason, no image augmentation was used.

**Interpretability** In Fig. 6, the 10 components with the highest positive reasoning probabilities for three exemplary classes are presented. After training the components in the feature space, the input representation of the components is determined by searching for the highest detection probability in the training set for the given component and cropping the corresponding image area in the input space. This method is similar to the approach from [7]. In general, the components with a high positive reasoning probability (above the initialization bound of $z$) are found to be conceptually meaningful for the respective class. Further investigation of the components shows that the detection of the component with the second highest positive reasoning probability for the dalmatian class in an image also provides evidence in favor of the giant panda class. Similarly, the component with the fifth highest positive reasoning probability for the dalmatian class is also highly important for the classes hyena, snow leopard, and english setter while the component with the fifth highest

positive reasoning probability for the class `trolleybus` is also important for the class `trolley car`. Similar shared components can be found across many classes, which shows that the CBC is capable of learning complex class-independent structures.

Averaged across all classes a positive reasoning probability greater than $z$ was learned for $5.2 \pm 0.8$ components per class while a negative reasoning probability greater than $z$ was assigned to $2\,781.8 \pm 23.3$ out of $5\,000$ components. As can be seen in Fig. 6, in most cases the positive reasoning probabilities assigned to components are close to $1.00$. This includes components that were not initialized with a bias towards the class in question. For example, the component with the fifth highest positive reasoning probability for the `dalmatian` class was initially biased towards the `english setter` class. The ratio between the number of positive and negative reasoning components suggest that the model heavily relies on negative reasoning to establish a baseline for its classification decision. We hypothesize that in this higher dimensional setting with a large number of components positive reasoning is primarily utilized to fine tune the models classification decision after rough categorization by negative reasoning.

**Performance**    To evaluate the performance of CBCs, we compare both the accuracy and inference time to that of a CNN. The resulting CBC had an inference time of $(371 \pm 6)$ images / sec, similar to $(369 \pm 2)$ images / sec of a normal ResNet-50 with global average pooling and fully-connected layer. This shows that the CBC generates no significant computational overhead. The top-5 validation accuracy of $82.4\%$ is on par with earlier CNN generations such as AlexNet with $82.8\%$ [48]. Note that the used CBC had a non-trainable feature extractor and no parameter tuning was performed. We are confident that the accuracy of CBCs on IMAGENET can be improved with further studies. The CBC was evaluated using one NVIDIA Tesla V100 32 GB GPU.

# 5    Conclusion and outlook

In this paper, we have presented a probabilistic classification model called classification-by-components network together with several possible realizations. Boiling down to the essential change we made, this is the definition of a probabilistic framework for the final and penultimate layer of a NN. The detection probability layer is an extension of a convolution layer with the requirement to measure the detection of convolutional filters called components, expressed in probabilities. Moreover, the final reasoning layer is still affine but follows a special implicit constraint defined by the probability model. The overall output is a probability value for each class without any artificial squashing. Independently of the feature extractor used in the CBC, we can always take advantage of this relation during inference by redefining the network to a single feedforward NN such that almost no computational overhead is created. This is shown in the experiment on IMAGENET.

Depending on the training setup, the method inherently contains a lot of different interpretation properties which are all founded on the new probability framework. As shown in the MNIST experiments with Siamese architectures, the method can produce human understandable components and is able to converge to the BMPP without any explicit regularization. Additionally, we have shown that the models can answer questions about the classification decision by an experiment with patch components on MNIST. More precisely, the model shows what causes the failure on an adversarial example. The conclusion drawn here supports the recently published results in [49]. A drawback of the Siamese architecture is the training overhead and the potential introduction of a lot of parameters due to components in the input space. In the non Siamese training, CBCs have almost no downsides to NNs. To be able to use all the presented interpretation techniques, the back projection strategy presented in [7] can be applied, as we have shown on IMAGENET. The evaluation on IMAGENET also showed that CBCs are capable of learning high dimensional components that can be utilized by multiple classes. Investigation of these shared components can provide additional insight into the model's classification approach. The heatmap visualizations are always applicable and extend the familiar CAM method by the option to visualize disagreement.

The CBC is a promising new method for classification and motivates further research. An initial robustness evaluation and the use of the class hypothesis possibility vectors for outlier detection show promising results, see supplementary material in Sec. E.2.4. Nevertheless, the following remain unanswered: What are proper regularizations for $\alpha_{c,i}$? What are more suitable detection probability functions? What are the advantages of the explicit injection of knowledge into the network in the form of trainable or non-trainable components, as we partly applied in the IMAGENET experiment?

**Acknowledgements**

We would like to thank Peter Schlicht and Jacek Bodziony from Volkswagen AG, Jensun Ravichandran from the University of Applied Sciences Mittweida, and Frank-Michael Schleif from the University of Applied Sciences Würzburg-Schweinfurt for their valuable input on previous versions of the manuscript. We would also like to thank the whole team at the Innovation Campus from Porsche AG, especially Emilio Oldenziel, Philip Elspas, Mathis Brosowsky, Simon Isele, Simon Mates, and Sebastian Söhner for their continued support and input. Lastly, we would like to thank our attentive anonymous reviewers whose comments have greatly improved this manuscript.

## Footnotes

[2]Note that the idea to explicitly model the state that a component does not contribute and avoid the general probabilistic approach $r_{c,k}^+ = 1 - r_{c,k}^-$ is related to the DEMPSTER–SHAFER theory of evidence [13].

[3]In contrast to prototypes, components are *not* class-dependent.

[4]The idea is to learn patches of: four quarters of a circle plus two diagonal, horizontal, and vertical lines.

[5]The symbol "$\circ$" denotes the Hadamard product (element-wise multiplication).

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
