[Supplementary Material]

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

[6]We experimented with networks where the priors $P(k)$ were kept as trainable parameters. The assumption that they have to add up to one was modeled by a softmax squashing. We found no real advantage to keep these parameters trainable. Nevertheless, to do so might be useful when training sparse models or performing a pruning of the components after the training by thresholding over the learned priors.

[7]https://keras.io/

[8]https://www.tensorflow.org/

[9]This is a suboptimal initialization of the components as it does not include any prior knowledge obtained from having access to the dataset before training. Using class-wise means or (partial) samples of the dataset might lead to earlier convergence.

[10]In [52], the application of Euclidean normalizations in NNs is studied.

[11]A *feature map* is the output produced by one filter of a convolutional layer. Moreover, the *feature map index* refers to the filter index.

[12]For further research we want to study rejection decisions regarding the measure "probability gap times predicted class probability".

[13]It is possible to upsample the reasoning stack first to get a higher resolution in the target image. We upsampled first to a size of $v_x' \times h_x' \times \#\mathcal{K}$ which results in a target image of $v_x \times h_x$.

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

# Supplementary material

## A   List of mathematical symbols

$\mathbf{0}$      zero vector

$\mathbf{1}$      one vector

$A$      binary random variable for agreement

$A^{\pm}, \overline{A}^{\pm}$      binary random variable for positive and negative (dis)agreement

$\mathbf{a}_c, \mathbf{b}_c$      coding vectors of the reasoning possibility vectors / stacks for class $c$ during training

$\mathfrak{a}, \mathfrak{a}_p, \mathfrak{a}_p^*$      an adversarial attack, an adversarial $L^p$-attack, worst-case adversarial $L^p$-attack

acc-$\mathfrak{a}$      adversarial threshold accuracy on the dataset $\mathcal{T}$

acc-$\mathfrak{a}_p^*$      worst-case adversarial threshold accuracy of $L^p$-attacks on the dataset $\mathcal{T}$

$\alpha_{c,i}, \alpha_{c,i,j}$      class-wise pixel probabilities

$\tilde{\alpha}_{c,i}, \tilde{\alpha}_{c,i,j}$      encoded class-wise pixel probabilities as elements of $\mathbb{R}$

$\beta$      margin parameter of margin loss

$\mathcal{C}$      set of all class labels $\{1, ..., \#\mathcal{C}\}$

$\#\mathcal{C}$      number of classes

$c$      class label $c \in \mathcal{C}$ or indicator variable of the class

$D$      binary random variable for detection

$\mathbf{d}(\mathbf{x})$      detection possibility vector / stack given $\mathbf{x}$

$d_k(\mathbf{x})$      detection probability function for component $\boldsymbol{\kappa}_k$ given $\mathbf{x}$

$\delta_{\mathfrak{a}}(\mathbf{x}, y)$      adversarial distance of attack $\mathfrak{a}$ on the sample $(\mathbf{x}, y)$

median-$\delta_{\mathfrak{a}}$      median adversarial distance of attack $\mathfrak{a}$ on the dataset $\mathcal{T}$

median-$\delta_p^*$      worst-case median adversarial distance of $L^p$-attacks on the dataset $\mathcal{T}$

$\mathbf{f}$      feature extractor

$h_d$      horizontal spatial dimension of a detection possibility stack

$h_{\kappa}$      horizontal spatial dimension of a component $\boldsymbol{\kappa}$

$h_{\kappa}'$      horizontal spatial dimension of a component $\boldsymbol{\kappa}$ after feature extraction

$h_p$      horizontal spatial dimension of a class hypothesis possibility stack

$h_r$      horizontal spatial dimension of a reasoning stack

$h_x$      horizontal spatial dimension of input $\mathbf{x}$

$h_x'$      horizontal spatial dimension of input $\mathbf{x}$ after feature extraction

$I$      binary random variable for importance

$\mathcal{K}$      set of all components $\{\boldsymbol{\kappa}_1, ..., \boldsymbol{\kappa}_{\#\mathcal{K}}\}$

$\#\mathcal{K}$      number of components

$k$      indicator variable of the component

$\boldsymbol{\kappa}$      component variable

$L^p$      $p$-norm

$l(\mathbf{x}, y)$      loss function given $\mathbf{x}$ and the corresponding class label $y$

| | |
|---|---|
| $m_\kappa$ | dimension of a component $\kappa$ after feature extraction |
| $m_x$ | dimension of the input $\mathbf{x}$ after feature extraction |
| $n_0$ | dimension of the receptive field |
| $n_\kappa$ | dimension of a component $\kappa$ |
| $n_x$ | dimension of the input $\mathbf{x}$ |
| $\mathbf{p}(\mathbf{x})$ | class hypothesis possibility vector given $\mathbf{x}$ |
| $p_c(\mathbf{x})$ | class hypothesis probability for class $c$ given $\mathbf{x}$ |
| $\phi(x)$ | squashing function |
| $R$ | binary random variable for reasoning by detection |
| $r_{c,k}^+, r_{c,k}^0, r_{c,k}^-$ | positive, indefinite and negative reasoning probability of class $c$ and component $\kappa_k$ |
| $\mathbf{r}_c^+, \mathbf{r}_c^0, \mathbf{r}_c^-$ | positive, indefinite and negative reasoning possibility vector / stack for class $c$ |
| $\bar{\mathbf{r}}_c^+, \bar{\mathbf{r}}_c^-$ | positive and negative effective reasoning possibility vector / stack for class $c$ |
| $\sigma$ | scaling parameter in the negative exponential of the Euclidean distance |
| $T, T_c$ | probability tree diagram, sub-tree of the probability tree diagram regarding class $c$ |
| $\mathcal{T}$ | dataset |
| $t_p, t_0, t_2, t_\infty$ | adversarial threshold parameters for adversarial threshold accuracies |
| $t_y, t_\beta$ | prediction and margin reject threshold parameter |
| $\boldsymbol{\theta}$ | trainable parameters of the feature extractor $\mathbf{f}$ |
| $v_d$ | vertical spatial dimension of a detection possibility stack |
| $v_\kappa$ | vertical spatial dimension of a component $\kappa$ |
| $v_\kappa'$ | vertical spatial dimension of a component $\kappa$ after feature extraction |
| $v_p$ | vertical spatial dimension of a class hypothesis possibility stack |
| $v_r$ | vertical spatial dimension of a reasoning stack |
| $v_x$ | vertical spatial dimension of input $\mathbf{x}$ |
| $v_x'$ | vertical spatial dimension of input $\mathbf{x}$ after feature extraction |
| $\mathbf{x}$ | input variable |
| $\tilde{\mathbf{x}}$ | an adversarial example of $\mathbf{x}$ |
| $y$ | class label of $\mathbf{x}$ |
| $\mathbf{z}$ | an arbitrary detection possibility vector |

## B Mathematical derivation of the class probabilities

First, the class hypothesis probabilities $p_c(\mathbf{x})$ of Eq. (1) are derived and their relation to a probability tree diagram $T$ is presented. Afterwards, we show how the class hypothesis *possibility* vector $\mathbf{p}(\mathbf{x})$ can be transformed into a class *probability* vector. The latter is required to use loss functions such as the cross entropy loss.

## B.1 Derivation of the class hypothesis probabilities

The proposed framework relies on a probabilistic model based on a probability tree diagram $T$. This tree $T$ can be decomposed into sub-trees $T_c$ for each class $c$ with the prior class probability $P(c)$ on the starting edge. Such a sub-tree is depicted in Fig. 2. For better readability, we dropped the variable $c$ in the probability tree diagram in Fig. 2, knowing that all probabilities except the ones on the bottom level, depend on $c$. We keep the class variable in the mathematical derivation below.

The whole probability tree diagram is modeled over five random variables:

- $c \in \{1, ..., \#\mathcal{C}\}$, indicator variable of the class;
- $k \in \{1, ..., \#\mathcal{K}\}$, indicator variable of the component;
- $I$, binary random variable for importance;
- $R$, binary random variable for reasoning by detection;
- $D$, binary random variable for detection.

The probabilities in the probability tree diagram are interpreted in the following way:

- $P(k)$, prior probability that the $k$-th component occurs;
- $P(I|k, c)$ and $P(\overline{I}|k, c)$, probability that the $k$-th component is important / not important for the class $c$;
- $P(R|k, c)$ and $P(\overline{R}|k, c)$, probability that the $k$-th component has to be detected / not detected for the class $c$;
- $P(D|k, \mathbf{x})$ and $P(\overline{D}|k, \mathbf{x})$, probability that the $k$-th component is detected / not detected in the input $\mathbf{x}$.

We compute the class hypothesis probability $p_c(\mathbf{x})$ regarding the paths of agreement $A$ under the condition of importance $I$. An agreement $A$ is a path in the probability tree diagram where either a component is detected $(D)$ and requires reasoning by detection $(R)$, or a component is not detected $(\overline{D})$ and requires reasoning by no detection $(\overline{R})$. The paths of agreement are marked with solid lines in Fig. 2. Hence, we model $p_c(\mathbf{x})$ by $P(A|I, \mathbf{x}, c)$:

$$P(A|I, \mathbf{x}, c) = \frac{P(A, I|\mathbf{x}, c)}{P(I|c)},$$
$$= \frac{\sum_k P(A, I|k, \mathbf{x}, c) P(k)}{\sum_k P(I|k, c) P(k)}.$$

Using the definition of $A$, the final equation is

$$P(A|I, \mathbf{x}, c) = \frac{\sum_k \left( P(R, D, I|k, \mathbf{x}, c) + P(\overline{R}, \overline{D}, I|k, \mathbf{x}, c) \right) P(k)}{\sum_k \left( 1 - P(\overline{I}|k, c) \right) P(k)},$$
$$= \frac{\sum_k \left( P(R|k, c) P(D|k, \mathbf{x}) + P(\overline{R}|k, c) P(\overline{D}|k, \mathbf{x}) \right) P(I|k, c) P(k)}{\sum_k \left( 1 - P(\overline{I}|k, c) \right) P(k)}. \quad (3)$$

During the training the reasoning probabilities $r_{c,k}^+$, $r_{c,k}^0$, and $r_{c,k}^-$ are learned. They are defined as:

- Positive reasoning: $r_{c,k}^+ = P(I|k, c) P(R|k, c)$ ... the probability that the $k$-th component is important and must be detected to support the class hypothesis $c$.
- Negative reasoning: $r_{c,k}^- = P(I|k, c) P(\overline{R}|k, c)$ ... the probability that the $k$-th component is important and must *not* be detected to support the class hypothesis $c$.
- Indefinite reasoning: $r_{c,k}^0 = P(\overline{I}|k, c)$ ... the probability that the $k$-th component is not important for the class hypothesis $c$.

Adding the relations $P(D|k, \mathbf{x}) = d_k(\mathbf{x})$ and $P(\overline{D}|k, \mathbf{x}) = 1 - d_k(\mathbf{x})$ and assuming that $P(k) = \frac{1}{\#\mathcal{K}}$ leads to[6]

$$P(A|I, \mathbf{x}, c) = \frac{\sum_k \left( r_{c,k}^+ d_k(\mathbf{x}) + r_{c,k}^- (1 - d_k(\mathbf{x})) \right)}{\sum_k \left( 1 - r_{c,k}^- \right)}. \tag{4}$$

The statement $r_{c,k}^+ + r_{c,k}^0 + r_{c,k}^- = 1$ is obvious, replacing the summations with the derived probabilities:

$$P(R|k, c) P(I|k) + P(\overline{R}|k, c) P(I|k, c) + P(\overline{I}|k, c) = 1,$$
$$\left( P(R|k, c) + P(\overline{R}|k, c) \right) P(I|k, c) + P(\overline{I}|k, c) = 1.$$

A reformulation of Eq. (4) with matrix calculus yields Eq. (1).

### B.2 Derivation of the class probability vector

For some applications it is useful to have a class probability vector instead of a class hypothesis possibility vector. To achieve this, the class hypothesis probabilities have to be normalized. The normalization into a class probability vector can be achieved by the derivation of the probability for a class $c$ under the condition of an agreement $A$ and importance $I$

$$
\begin{aligned}
P(c|A, I, \mathbf{x}) &= \frac{P(c, A, I, \mathbf{x})}{P(A, I, \mathbf{x})}, \\
&= \frac{P(c, A|I, \mathbf{x})}{P(A|I, \mathbf{x})}, \\
&= \frac{P(A|I, \mathbf{x}, c) P(c)}{\sum_{c' \in \mathcal{C}} P(A|I, \mathbf{x}, c') P(c')}, \\
&= \frac{p_c(\mathbf{x}) P(c)}{\sum_{c' \in \mathcal{C}} p_{c'}(\mathbf{x}) P(c')}, \tag{5}
\end{aligned}
$$

where $P(c)$ is the prior probability of the class $c$. Hence, the transformation is obtained by dividing class-wise by the sum of all class hypothesis probabilities.

## C  Partial spatial reasoning

In the following, the most generic model of the presented reasoning process is described. Each reasoning process (spatial reasoning or reasoning over full-size components) defined in the main paper can be derived in terms of this generic model. In the initial description of spatial reasoning, it is assumed that the spatial dimension of the reasoning possibility stack $v_r$ is equivalent to the spatial dimension of the detection possibility stack $v_d$. In the general case, we relax this assumption to $v_r \leq v_d$. For example in case of image inputs, the detection possibility stack has a dimension of $v_d \times h_d \times \#\mathcal{K}$ and the reasoning possibility stack the dimension of $v_r \times h_r \times \#\mathcal{K}$ with $v_d \geq v_r$ and $h_d \geq h_r$. Now, instead of computing just one class hypothesis probability $p_c(\mathbf{x})$ as in spatial reasoning, we slide the spatial reasoning process of Fig. 3 with the smaller reasoning possibility stack over the detection possibility stack. This results in a class hypothesis probability map of size $v_p \times h_p$. Additionally, we collect the probabilities of all classes into a stack of size $v_p \times h_p \times \#\mathcal{C}$. As a final step we downsample the possibility stack to a class hypothesis possibility vector, e. g. by applying global max pooling.

The idea behind this approach is that we search at different positions for a match of the learned class DP in the extracted DP. Doing so, we can handle local shifts and, hence, detect objects at different positions in general. In this case, the reasoning process is only applied over a part of the detection possibility stack, thus, we call it *partial spatial reasoning*. This operation can be efficiently implemented as the reasoning process Eq. (1) is equivalent to an affine transformation.

# D   Training of classification-by-components networks

**Training of the components**   If defined as trainable parameters in the input space, it is required to constrain the components $\kappa_k$ to remain in this space. This is achieved by performing a form of projected gradient descent learning, where the components are clipped back into the correct range after each update. If the components are defined as trainable parameters in the feature space, this is not required and they can be trained over $\mathbb{R}$.

The projected gradient descent learning is used during the experiments on MNIST, CIFAR-10, and GTSRB, where the input space is defined over the interval $[0, 1]$. For the experiment on IMAGENET, no additional constraint is needed as the components are defined in the feature space.

**Training of the reasoning possibility vectors**   One main condition in the CBCs is that the trainable reasoning vectors $\mathbf{r}_c^+$, $\mathbf{r}_c^0$, and $\mathbf{r}_c^-$ are elements of $[0, 1]^{\#\mathcal{K}}$ and add up to one, i.e., $\mathbf{r}_c^+ + \mathbf{r}_c^0 + \mathbf{r}_c^- = \mathbf{1}$. This condition is preserved by encoding the three reasoning vectors for each class into two vectors that can be learned. These two vectors are denoted as $\mathbf{a}_c$ and $\mathbf{b}_c$ and are defined to be elements of $[0, 1]^{\#\mathcal{K}}$. The decoding into the reasoning vectors is defined by

$$\mathbf{r}_c^+ = \mathbf{a}_c,$$
$$\mathbf{r}_c^- = (\mathbf{1} - \mathbf{a}_c) \circ \mathbf{b}_c,$$
$$\mathbf{r}_c^0 = \mathbf{1} - \mathbf{a}_c - (\mathbf{1} - \mathbf{a}_c) \circ \mathbf{b}_c.$$

During the training of a network the parameters $\mathbf{a}_c$ and $\mathbf{b}_c$ are constrained to $[0, 1]^{\#\mathcal{K}}$ by clipping values outside the interval after each update which can be seen again as a form of projected gradient descent learning. To avoid the computation of $\mathbf{r}_c^0$ we can substitute $\mathbf{1} - \mathbf{r}_c^0$ in Eq. (1) by $\mathbf{r}_c^+ + \mathbf{r}_c^-$.

**Training of the pixel probabilities**   In the CBCs with spatial reasoning, the class-wise pixel probabilities $\alpha_{c,i}$ are potentially trainable parameters. By definition, it must hold that $\alpha_{c,i} \in [0, 1]$ and $\sum_{i,j} \alpha_{c,i} = 1$. This is preserved by learning encoded parameters $\tilde{\alpha}_{c,i} \in \mathbb{R}$ that can be decoded using a softmax activation

$$\alpha_{c,i} = \frac{\exp\left(\tilde{\alpha}_{c,i}\right)}{\sum_{i'} \exp\left(\tilde{\alpha}_{c,i'}\right)}$$

to get the pixel probabilities $\alpha_{c,i}$.

**End-to-end learning of a CBC**   An important observation is that the components are in general *not* equal to any elements of the dataset. This fact is important for the training of CBCs that use a Siamese setup to extract features from both the inputs and the components. Particularly, one path of the Siamese feature extractor processes the input samples and the other processes the components. To solve the classification problem, it is important that the feature extractor is only updated to extract characterizing features from dataset samples and not from the non-dataset samples (the components). Hence, the gradient backflow to the parameters of the feature extractor path, which is responsible for extracting features from the components, has to be stopped. The gradient backflow through this path is only used for updating the components if they are trainable.

While this is a slight deviation from the usual training procedure of a Siamese network, it improves the training of interpretable components significantly and is theoretically grounded.

# E   Extended evaluation

In this section, we present an extended evaluation on MNIST, GTSRB and CIFAR-10. Before we start, we describe general settings of the CBCs which were used throughout all experiments. After that, we start with MNIST and present the results of the ablation study and some first robustness evaluation results. We continue with GTSRB where we show how a CBC performs on the physical adversarial images from [50]. Finally, we present results on CIFAR-10.

### E.1 General training setup and network setting

If not stated differently, all networks were trained with the following setting on a single NVIDIA Tesla V100 32 GB GPU using KERAS[7] with TENSORFLOW[8] backend. The input samples were always normalized to the input space defined over $[0, 1]$. If components / prototypes are defined in the input space, then they are constrained to this space. All networks were trained for 150 epochs with the default Adam optimizer from KERAS with an initial learning rate of 0.003. The learning rate was automatically decreased by a factor of 0.9 when the validation loss did not improve for five epochs. We used a batch size of 128 and data augmentation consisting of $\pm 2$ pixels random shifts and $\pm 15°$ random rotations. The loss function of the CBCs was the introduced margin loss with a margin of $\beta = 0.3$. We trained the whole networks from scratch and initialized the reasoning probabilities with $[0, 1]$ and the components with $[0.45, 0.55]$ uniform random noise.[9] The detection probability function is the cosine similarity with ReLU activation to clip the negative part.

Depending on the performed experiment, different CNNs are used as feature extractor for the CBC. With the exception of the CNN feature extractor used in the IMAGENET experiment, all feature extractors have a few settings in common. Specifically, the convolutional filters in the feature extractor are constrained to have an Euclidean norm of one and are activated by the Swish activation function. Additionally, the feature extractors do not contain any batch normalization. This setting is chosen based on the results of the ablation study presented in Sec. E.2.2. From here on, we will denote this setting of a CNN feature extractor in combination with the margin loss and cosine similarity detection probability function as the *standard setting*. In contrast, when a CNN feature extractor is used that is more in line with common CNN architectures (no convolutional filter constraint, ReLU activation and batch normalization) we will refer to it as the *non-standard setting*. CBCs with the non-standard setting still use the margin loss and cosine similarity detection probability function, if not stated differently.

All networks were trained at least three times. We report the mean and standard deviation of the test accuracy and give notice when a run diverged. Moreover, we do not report early stop accuracies. In addition, we present the average probability gap of a CBC over the test set. The *probability gap* is defined as

$$\max \left\{ p_c \left( \mathbf{x} \right) | c \in \mathcal{C} \right\} - \max \left\{ p_{c'} \left( \mathbf{x} \right) | c' \in \mathcal{C}, c' \neq \arg\max \left\{ p_c \left( \mathbf{x} \right) | c \in \mathcal{C} \right\} \right\}.$$

This value is different from the probability gap (margin) optimized by the margin loss, as the predicted class is not necessarily equivalent to the correct class.

**A note about the detection probability function**   In parallel to the cosine similarity as detection probability function, in a couple of experiments, we studied the use of the Euclidean distance activated by the negative exponential. As stated in [16], the Euclidean distance can be trained in deep NNs if one takes the distribution of the distance (e. g. the $\chi^2$ statistics if the requirements for this distribution are fulfilled) into account. Otherwise, the Euclidean distance together with the negative exponential leads to the vanishing gradients problem. In the evaluations, we tackled this problem by training a scaling parameter $\sigma \in \mathbb{R}_{>0}$ such that the detection probability function becomes

$$d_k \left( \mathbf{x} \right) = \exp \left( -\frac{\| \mathbf{f} \left( \mathbf{x} \right) - \mathbf{f} \left( \boldsymbol{\kappa}_k \right) \|_2^2}{\sigma} \right). \tag{6}$$

Using this approach with the trainable parameter $\sigma$, a CBC with the Euclidean distance was able to achieve similar results to a CBC with the cosine similarity. However, the stability of the training was sensitive to the initialization of $\sigma$ and was prone to diverge. We see this as an indicator for the detection probability function being crucial for the success of the CBCs and therefore consider it as one of the main focus points of further research.

Figure 7: Learned reasoning process of a CBC without feature extractor, 10 components and trained over a margin $\beta = 0.3$ on MNIST. *Top row:* The learned components. *Bottom row:* The learned reasoning probabilities collected in reasoning matrices. The class is indicated by the MNIST digit below. The top row corresponds to $r_{c,k}^+$, middle row to $r_{c,k}^0$, and bottom row to $r_{c,k}^-$. White squares depict a probability of one and black squares of zero.

## E.2 MNIST

In this section we present the full evaluation of CBCs with experiments performed on MNIST. The first experiment evaluates the performance of a CBC without a feature extractor on top and compares it to a traditional prototype-based classifier. Following this, we present the results of the ablation study and discuss the impact of certain parameters on the CBC performance. Similar to Sec. 4.1, we use the results of the ablation study to perform a set of experiments with a CBC with feature extractor and a varying number of components. This set of experiments leads to an observation regarding the relation between the interpretability of components and the robustness of the network against adversarial attacks, which we discuss afterwards. Finally, we close this section by presenting additional visualizations and provide an extended description of the construction process of the presented visualizations for patch component CBCs.

### E.2.1 CBCs without a feature extractor

**Experiment description** The purpose of this experiment is to compare the CBC to a prototype-based classifier. Additionally, we want to show that the reasoning process is a natural extension to the BMPP employed by them. For a fair comparison, the identity mapping is used as feature extractor of the CBC. Therefore, the detection probability function computes the dissimilarity directly in the input space (as in most prototype-based classifiers). One of the design goals for the CBC is that the BMPP is a valid solution of the classification process. Therefore, we use an architecture with 10 full-size components, which converges to the BMPP if we train for a robust classification (high enough margin $\beta$ in the margin loss). Additionally, we show that the CBC can discover an alternative to the BMPP if we allow the classification to be based on weak decisions. This is done by lowering the margin $\beta$ in the margin loss.

The prototype-based classifier that we compare to the CBC is the Generalized Learning Vector Quantization (GLVQ) method [14] which is a supervised, end-to-end trainable classifier. As baseline, we use GLVQ with one prototype per class and squared Euclidean distance.

**Accuracy results** In terms of accuracy the CBC performs slightly better than GLVQ with a test accuracy of $(83.5 \pm 0.12)\%$ against $(81.7 \pm 0.22)\%$. If we reduce the margin $\beta$ to $0.1$ the accuracy increases to $(89.5 \pm 0.12)\%$ and clearly outperforms the learned GLVQ model. We hypothesize that the reason for this is that with a smaller margin the network has the freedom to classify by weak decisions. The combined optimization of forcing a margin of $0.3$ and high accuracies might result in a trade-off between accuracy and the robustness of the decisions. On the contrary, GLVQ is by design a hypothesis margin maximizer [15]. Thus it is optimizing for a maximum averaged relative margin over all samples and, therefore, forces the model to robust decisions.

Empirical evidence supporting this weak decision hypothesis can be found by evaluating the average probability gap of the CBC classifications. While for a margin of $0.3$ this gap is $0.16 \pm 0.11$, it is only $0.1 \pm 0.05$ for a margin of $0.1$. For this evaluation only the correct classified images were taken into account and, therefore, the direct comparison to the optimization margin can be made.

**Interpretation of the reasoning with a margin of 0.3** In Fig. 7 we depict the learned reasoning process with a margin of $\beta = 0.3$. The network converges consistently towards the BMPP and

Figure 8: Class-specific prototypes learned by GLVQ. The class is indicated by the MNIST digit below.

Figure 9: Learned reasoning process of a CBC without a feature extractor, 10 components and trained over a margin $\beta = 0.1$ on MNIST. *Top row:* The learned components. *Bottom row:* The learned reasoning probabilities collected in reasoning matrices. The class is indicated by the MNIST digit below. The top row corresponds to $r_{c,k}^+$, middle row to $r_{c,k}^0$, and bottom row to $r_{c,k}^-$. White squares depict a probability of one and black squares of zero.

turns the class-independent components into prototypical (class-specific) components. The learned components can directly be recognized as class-specific components as they resemble digit shapes. The BMPP is learned if the indefinite reasoning probabilities for all components, except for one, are close to one and if the remaining component is used for positive reasoning. Only for the class 1 the CBC performs a slightly negative reasoning over the component f which can be seen at the grayish negative reasoning probability. It is a remarkable observation that the CBC converges consistently towards the BMPP, without any regularization or constraint over the reasoning process. This behavior shows that the BMPP is a stable solution for a robust classification process. The learned components are similar to the learned prototypes of the GLVQ model in Fig. 8. For example, compare the strong component of class 5 with the learned GLVQ prototype. This provides additional evidence for the claim that a CBC can act similar to a prototype-based classifier, but only if this is the best classification principle for the task. The differences between the learned prototypes and components are the result of the different similarity measures. If we switch to Eq. (6) as detection probability function in the CBC, then the learned components start to become indistinguishable to the learned prototypes.

**Interpretation of the reasoning with a margin of 0.1**    In contrast to the learned reasoning process with a margin of $\beta = 0.3$, the reasoning process over a low margin is complementary to the BMPP, see Fig. 9. Almost all the decisions are based on strong negative reasoning over one component. Only a few classes show weak positive reasoning over a few components. Hence, the learned components can be interpreted as negative examples of the classes: To make a correct classification of a given image, the strong (negative) component of the correct class should have the lowest detection probability in the image. The interpretation of negative samples is hard as it is not a human intuitive reasoning process.

Albeit a bit less intuitive, it is still possible to understand the classification decision. Consider the reasoning process of the class 0, which relies on strong negative reasoning over component h. This component has a bright dot in the middle and the rest remains almost black. It is not hard to see that this is indeed a negative example for the class 0 as a zero is the only digit which has no stroke in the middle. The same interpretation can be applied for class 1 and the respective component. A more complex interpretation is the reasoning of the class 7 and negative component i. The black top horizontal line indicates a unique property of the seven and a five: the horizontal top stroke. To avoid confusions to a five it strongly highlights the region where a five has the transition from the vertical stroke to the arc – which means this should not be present. The property, that the classification is sometimes based on just finding unique parts and does not understand the whole image, was in general observed in a couple of low margin experiments, also beyond MNIST.

Figure 10: Example components for the visual evaluation of the interpretability. We excluded examples of the score category five as these are assumed as real input images.

## E.2.2 Ablation study

As described in Sec. 4.1, we found a strong variation within the interpretability of the components. The goal of this study was to find out how different settings of a CBC affect the interpretability of its components.

**Experimental setup** The following 4-layer CNN is used as feature extractor for the evaluation:

1. Convolution: 32 filters, kernel size 3×3, stride 1×1, bias, no padding;
2. Convolution: 64 filters, kernel size 3×3, stride 1×1, bias, no padding;
3. Max pooling: pool size and stride 2×2;
4. Convolution: 64 filters, kernel size 3×3, stride 1×1, bias, no padding;
5. Convolution: 128 filters, kernel size 3×3, stride 1×1, bias, no padding;
6. Max pooling: pool size and stride 2×2.

The reasoning is defined with 10 full-size components. We experimented with all possible combinations of the following parameters and their configurations (`<parameter description: configuration-1, configuration-2, ...>`):

- activation function of convolutional layers: ReLU, Swish;
- constraint of convolutional filters (kernels) to Euclidean norm one: true, false;
- application of batch normalization after the first max pooling layer: true, false;
- activation of the cosine similarity to provide the detection probability function: $\mathrm{ReLU}\,(x)$, $(\mathrm{ReLU}\,(x))^2$;
- squashing function $\phi$ in Eq. (2): Exponential Linear Unit (ELU) [51] function, margin loss with $\beta = 0.1$, margin loss with $\beta = 0.3$, margin loss with $\beta = 0.5$, margin loss with $\beta = 0.7$, margin loss with $\beta = 0.9$.

**Evaluation** After the networks were trained, we extracted the final components for each combination. We asked 10 human experts to give scores to the interpretability of components by visual examination. For each combination, we asked for one score for all resulting components. The score values were described according to the following definitions and accompanied by the examples in Fig. 10:

**Score of 0:** The images resemble unstructured noise with no visible digit shapes.

**Score of 1:** The images show something resembling a digit, but the image contains a lot of noise.

**Score of 2:** The images show something resembling a digit, but might contain artifacts of other digits.

**Score of 3:** The images show real looking digits, but the background contains a lot of noise.

**Score of 4:** The images show real looking digits and background contains close to no noise.

**Score of 5:** The images are indistinguishable from real inputs.

The final interpretability score is computed as the average. Additionally to the interpretation score, we collected some statistics over the training / test accuracy / loss.

Figure 11: Impact of the squashing function $\phi$ on the interpretability score and the test accuracy.

**Results** Since the overall goal of the ablation study is to find a CBC setting where the components are learned to be human interpretable, the first evaluation criterion is the interpretability score. Two of the combinations scored an interpretability score of 3.8 (the highest score that was given to a model by a single expert was 4). The difference between these two models is the loss function in use, see Fig. 11. The first model was trained with ELU function for $\phi$ whereas the other model was trained over the margin loss with a margin of $\beta = 0.3$. To decide which model performs better, we compared their final test accuracies. The model with the margin loss performed significantly better: $(99.27 \pm 0.1)\,\%$ average test accuracy with margin loss and $(97.86 \pm 0.19)\,\%$ with ELU function. Hence, the combination of this point was chosen to be the *standard setting* for CBCs throughout all experiments. Interestingly, the whole evaluation showed that the accuracy is not correlated with the interpretability score, see Fig. 11. Hence, a model with non-interpretable components was able to reach high accuracies.

In Fig. 12 we visualize the effect of the other parameters. There is no clear trend regarding the cosine similarity activation but a clear trend with respect to the other parameters. The defined standard setting clearly outperforms all others.

**Analysis of the results** In general, the Euclidean constraint seems to have the strongest impact on the interpretability, see Fig. 12. We observed the positive effect of this constraint during an experiment with convolutional filters predefined as Sobel operators. The idea behind this constraint is to make all the filters comparable and normalize them in such a way that each can distribute 100% of energy over their weights.[10] Later we realized that in the non-standard setting (ReLU activation of convolutional layers, batch normalization, and no constraint), the activations seem to explode with increasing network depth of the feature extractor, see Fig. 13a. Note the different scales of the horizontal axis of the two plots. Independently to that, the CBC with the non-standard setting trained to an acceptable, but lower accuracy of $(98.86 \pm 0.26)\,\%$ compared to the standard setting. To understand how the network with the non-standard setting classifies, we draw some conclusions from the CBC architecture in use:

1. The cosine similarity is equivalent to a dot product (Euclidean inner product) after the two input vectors were normalized to an Euclidean norm of one.

2. If there are elements in the vector that are orders of magnitude larger than all other elements, then only the largest values will be nonzero after the normalization and only these values can add to a high output similarity.

3. After the normalization, a high detection probability of $d_k(\mathbf{x}) \approx 1$ implies $\frac{\mathbf{f}(\mathbf{x})}{\|\mathbf{f}(\mathbf{x})\|_2} \approx \frac{\mathbf{f}(\boldsymbol{\kappa}_k)}{\|\mathbf{f}(\boldsymbol{\kappa}_k)\|_2}$ and thus the input is similar to the $k$-th component.

(a) Impact of the activation of convolutional layers on the interpretability score and the test accuracy.

(b) Impact of the Euclidean constraint on the interpretability score and the test accuracy.

(c) Impact of batch normalization on the interpretability score and the test accuracy.

(d) Impact of the cosine similarity activation on the interpretability score and the test accuracy.

Figure 12: The results of the ablation study regarding the parameters convolutional activation, Euclidean constraint, batch normalization, and cosine similarity activation.

(a) CBC with non-standard setting.

(b) CBC with the standard setting.

Figure 13: The distribution of the input values (output activations of the feature extractor) to the detection probability function with the non-standard setting (a) and the standard setting (b) depicted as histograms. We evaluated the histograms over the training database.

(a) CBC with the non-standard setting.

(b) CBC with the standard setting.

Figure 14: Activations of the feature maps of the final convolutional layer in the feature extractor after Euclidean normalization. The activations are shown for the strong positive component of the class 4 for the non-standard setting (a) and the standard setting (b). The colors within one plot correspond to the spatial dimensions and show how different spatial dimensions within the $4\times4$ feature stack contribute. Above the plot, the corresponding learned component is depicted.

4. If the model classifies by the BMPP over the prototypical component $\boldsymbol{\kappa}_k$ for class $y$ and $p_y(\mathbf{x}) \approx 1$ is valid, then $d_k(\mathbf{x}) \approx 1$.

Both networks classify by the BMPP over prototypical components (see Fig. 14 for example components of both networks) with high output probabilities of $p_y(\mathbf{x}) \approx 1$ for correct classifications. The components of the CBC with non-standard setting are not human understandable while the components of the CBC with standard setting are. By conclusion 4 and 3 we know that after the normalization an input has to be approximately equivalent to the strong prototypical component. Moreover, by conclusion 2 we know that only nonzero values contribute to the classification decision.

In case of the non-standard setting, or in other words the setting most often used in CNNs, the nonzero values are those of high activation. Hence, the model outputs high activations to suppress small values. Moreover, the network relies only on a few significant features for each class, see Fig. 14a for an example activation pattern after normalization. The complete classification of a digit four is based on a few features of two different feature maps.[11] By this "trick" the network does not learn to extract useful features which can be used across the classes but instead amplifies a certain feature map for each class. The activation pattern of a certain class is not understandable as the components can not be interpreted.

With the standard setting the classification works differently. The network cannot overemphasize a few features as the filters are normalized by the Euclidean norm and can also not produce filters where all the weights are zero ("dead" filter). Hence, the network has to incorporate each filter and, therefore, learns highly discriminative features. If we consider the activation patterns across all classes, we observe that the network learns features that are used across multiple classes and that it classifies by combining several features, see the activation pattern after normalization of Fig. 14b. We hypothesize that the reason why the components become human interpretable, is that compared to the network with the non-standard setting, the network with the standard setting really learns to decode the whole digit into a high level representation, such as strokes and arcs. In contrast, the network with the non-standard setting just learns a unique pattern, for example the distribution of intensities.

Figure 15: Evolution of a component over the epochs of a 10 components CBC.

Figure 16: Learned reasoning process of a CBC with 10 components on MNIST. *Top row:* The learned components. *Bottom row:* The learned reasoning probabilities collected in reasoning matrices. The class is indicated by the MNIST digit below. The top row corresponds to $r_{c,k}^+$, middle row to $r_{c,k}^0$, and bottom row to $r_{c,k}^-$. White squares depict a probability of one and black squares of zero.

The different losses show no superior trend regarding a specific setting except that a too big margin in the margin loss is harmful for both interpretability and accuracy. From [53] and [54] we know that the optimization of a similarity (or metric) with a contrastive loss could lead the feature extractor to project all the points from one class to only one single point. This effect is known as collapsing dimensions and is the main reason why we do not apply losses like mean squared error or cross entropy as they rely on the optimization towards a one-hot label. This effect leads to highly non-linear regions in the learned mapping. We believe that the model needs "space" to distribute the feature vectors smoothly and well scattered to converge to the desired interpretability. A compromise between the optimization for correct classifications and the increase of the margin is the ELU squashing. It has a derivative of one for incorrectly classified samples and slowly scales down the updates of already correctly classified images to avoid a collapsing of dimensions. The ELU loss is an alternative to the margin loss and worked well throughout all experiments. However, it is always slightly below the margin loss in terms of accuracy. Moreover, we observe that, even when we are optimizing for a fixed margin of $\beta = 0.3$, the resulting margin is often much bigger, see Fig. 18. This is an indicator that the feature vectors are fairly smoothly distributed throughout the whole space.

So far, we have no explanation for why the batch normalization seems to be harmful for the interpretability. In contrast, we think that the improved behavior of Swish is due to the property that the function is differentiable everywhere and has almost always nonzero gradients.

### E.2.3   CBCs with feature extractor

In addition to the results over a 9 component CBC with feature extractor presented in Sec. 4.1.1, in this section we present additional results of a 10 and an 8 components model and show that the CBC can also deal with other numbers of components and still classifies well on MNIST. Afterwards, we show that the sparseness of a CBC (fewer components) does come with reduced performance as the models show a significantly smaller margin between the probabilities of the correct and the incorrect class with the highest probability. This indicates that the models start to classify by weaker decisions. The standard setting for CBCs is used for this experiment, making it equivalent to the 9 component CBC presented in Sec. 4.1.1 but with the different number of components.

**10 components CBC**   To give an impression on how the model learns the components, we present the evolution of a component starting from random noise in Fig. 15. After a few epochs the rough shapes of the modeled digits are visible and the remaining epochs are used for fine-tuning to remove background noise. In general, the training of the model is stable and converges consistently to high test accuracies of $(99.27 \pm 0.1)\,\%$. Interestingly, the model does not always converge clearly towards the BMPP with sharp prototypical components, as shown in Fig. 16. It occurred quite often that the model solves the reasoning for the class 1 with strong negative reasoning over one component. In this case 9 components were clearly class-specific in the human understandable

Figure 17: Learned reasoning process of a CBC with 8 components on MNIST. *Top row:* The learned components. *Bottom row:* The learned reasoning probabilities collected in reasoning matrices. The class is indicated by the MNIST digit below. The top row corresponds to $r_{c,k}^+$, middle row to $r_{c,k}^0$, and bottom row to $r_{c,k}^-$. White squares depict a probability of one and black squares of zero.

sense and the strong component for the class 1 remained in a state similar to the strong negative component in Fig. 4.

**8 components CBC**   If we train a CBC with 8 components and compare it to the models discussed before, we can observe a slight drop in accuracy to $(99.07 \pm 0.1)\,\%$. To classify MNIST with only 8 components, the model learns to reason negatively over a couple of components. Similar to before, the 1 is classified via negative reasoning. In addition to that, the model classifies the 2 with negative reasoning too. In contrast to all models before, the model classifies the 6 with a combination of positive and negative reasoning. Moreover, the learned classification principle becomes hard to understand. Despite some of the components being easy to interpret, it cannot be easily answered how the model uses component g for positive reasoning for class 8 and for negative reasoning for class 2.

**Comparison of the models**   The sparsity of the number of components comes with some downsides. First of all, the models become harder to interpret as shown in the comparison between the 10 component model in Fig. 16 and the 8 component model in Fig. 17. Secondly, the networks start to classify by weaker decisions if the number of components is reduced. This is, however, not directly noticeable in the accuracy as for all three models it is still above 99%.

All networks are optimized towards a probability margin of 0.3 between the correct and highest probable incorrect class with the margin loss. Despite that, the 10 components network has an average probability gap of $0.59 \pm 0.14$ over correctly classified images after training. This shows that even though we are optimizing only for a margin of 0.3 the networks can become more discriminative by themselves and distribute the probabilities through the whole range if possible, see Fig. 18a. The figure clearly highlights that incorrectly classified images are indicated by a high uncertainty as the probability gap is much smaller. This indicates that the runner-up class and the predicted class are almost equivalent. In other words, the model is not certain about its classification decision.

If we decrease the number of components in the CBCs, we observe that the probability gap of correctly classified images becomes smaller, see the shift of the distribution in Fig. 18b and Fig. 18c. At the same time the distribution of incorrectly classified images does not change that much. Overall, the densities clearly show that the decreased number of components lowers the credibility of the model's classification decision.

### E.2.4   Adversarial robustness and rejection

During the ablation study we found that for some settings of the feature extractor, the final trained components were more interpretable than for other settings. Interestingly, the network was able to achieve close to perfect classification with both interpretable and non-interpretable components. Given that the CBCs of the ablation study rely only on positive reasoning through the BMPP and that the components are defined in the same space as the input, we know that the components should be interpretable if $p_y(\mathbf{x}) \approx 1$ as it implies $d_k(\mathbf{x}) \approx 1$ for the prototypical component $\boldsymbol{\kappa}_k$. Hence, we hypothesize: If the CBC converged to the BMPP but the trained components are not interpretable although $p_y(\mathbf{x}) \approx 1$, then it is safe to assume that the network might be susceptible for adversarial attacks.

We performed a robustness evaluation to provide empirical evidence that this hypothesis is correct. Moreover, based on these results we show that the distribution of class hypothesis probabilities of

(a) 10 components CBC.

(b) 9 components CBC.

(c) 8 components CBC.

Figure 18: Distribution of the probability gap over incorrectly and correctly classified images by CBCs on MNIST with different numbers of components. We visualize discrete approximations of the continuous density functions.

Table 1: The results of the adversarial robustness evaluation. The attacks are grouped by the norm under which they are optimized, the boxes denote if the attack is either white or black box. For each model, we report the clean accuracy in %, the median-$\delta_{\mathfrak{a}}$ (left value) and acc-$\mathfrak{a}$ score (right value) in % for each attack, and the worst-case analysis over all $L^p$-attacks by presenting the median-$\delta_p^*$ (left value) and acc-$\mathfrak{a}_p^*$ score (right value) in %. Higher scores mean better robustness. For each attack the best median-$\delta_{\mathfrak{a}}$ is highlighted in bold.

| | | | CNN-0 | | CNN-4 | | CBC-0 | | CBC-4 | |
|---|---|---|---|---|---|---|---|---|---|---|
| | clean accuracy | | | 99.6 | | 99.6 | | 98.6 | | 99.4 |
| $L^2$ | DEEPFOOL | □ | 1.01 | 9.6 | 1.20 | 22.0 | 0.64 | 30.3 | **1.98** | 71.0 |
| | C&W | □ | 0.86 | 5.2 | 0.99 | 5.1 | 0.58 | 28.5 | **1.27** | 29.6 |
| | POINTWISE | ■ | 2.62 | 88.9 | 2.79 | 93.2 | 1.59 | 53.5 | **3.19** | 94.4 |
| | BOUNDARY | ■ | 1.12 | 19.2 | 1.29 | 28.2 | 0.32 | 3.0 | **1.69** | 66.0 |
| | **worst-case** | | 0.84 | 1.5 | 0.98 | 4.8 | 0.28 | 0.12 | **1.26** | 27.5 |
| $L^\infty$ | FGSM | □ | 0.19 | 21.0 | 0.16 | 12.6 | 0.11 | 25.2 | **0.24** | 30.6 |
| | DEEPFOOL | □ | 0.10 | 0.1 | 0.11 | 0.0 | 0.08 | 24.4 | **0.18** | 15.2 |
| | PGD | □ | 0.09 | 6.0 | 0.10 | 2.5 | 0.04 | 1.9 | **0.15** | 0.2 |
| | **worst-case** | | 0.09 | 0.0 | 0.09 | 0.0 | 0.04 | 1.9 | **0.15** | 0.02 |
| $L^0$ | POINTWISE | ■ | 5.0 | 3.3 | 5.0 | 5.9 | 2.0 | 0.1 | **7.0** | 16.0 |
| | S&P | ■ | 29.0 | 78.0 | 32.0 | 82.2 | 11.0 | 46.6 | **60.0** | 94.2 |
| | **worst-case** | | 5.0 | 3.3 | 5.0 | 5.9 | 2.0 | 0.01 | **7.0** | 16.0 |

a CBC can be used to detect when an image is manipulated. This property can be used for reject strategies as we show later. The same property is observed with GTSRB on real world adversarial examples, see Sec. E.3.

**Network setup** For the evaluation, two CBCs and two CNNs are used. The first model is equivalent to the CBC with the standard setting from the ablation study in Sec. E.2.2 with a high interpretability score of 3.8 and is called CBC-4 in the following. The second model is the equivalent CBC with the non-standard setting, which was also used previously in the ablation study in Sec. E.2.2. This model reaches an interpretability score of 0.4 and is called CBC-0.

The two CNNs are constructed by taking the feature extractor networks of the CBCs and adding two fully-connected layers on top. The fully-connected layers have 512 and 10 units respectively and are separated by a dropout of 0.5. The activation function for the first fully-connected layer is similar to the one chosen for the convolutional layers while for the final fully-connected layer the softmax activation function is used to produce a probability vector. Both networks were trained with the cross entropy loss. The other parameters of the two CNN baseline models were chosen such that CNN-0 was in line with CBC-0 and CNN-4 with CBC-4.

**Robustness evaluation** As the CBCs were not designed with a specific thread model in mind the robustness was evaluated over three different $L^p$-norms using both black and white box attacks: DEEPFOOL [55], CARLINI&WAGNER (C&W) [56], POINTWISE [57], FAST GRADIENT SIGN METHOD (FGSM) [8], BOUNDARY [46], PROJECTED GRADIENT DESCENT (PGD) [58], and SALT-AND-PEPPER noise attack (S&P) implemented in FOOLBOX [59]. We define an adversarial image $\tilde{\mathbf{x}}$ as a manipulated image of the input image $\mathbf{x}$ which is misclassified by the model. Moreover, we search for adversarial images such that $\|\mathbf{x} - \tilde{\mathbf{x}}\|_p$ becomes minimal. To evaluate the robustness, we tried to compute an adversary for *each* sample of the MNIST test dataset with *each* attack $\mathfrak{a}$ and computed the adversarial distance $\delta_{\mathfrak{a}}(\mathbf{x}, y)$. Given a sample $(\mathbf{x}, y)$ from the dataset $\mathcal{T}$ and a $L^p$-attack $\mathfrak{a}$, $\delta_{\mathfrak{a}}(\mathbf{x}, y)$ is defined as: **(1)** 0, if the data sample $\mathbf{x}$ is misclassified by the model; **(2)** $\|\tilde{\mathbf{x}} - \mathbf{x}\|_p$, if $\mathfrak{a}$ found an adversary $\tilde{\mathbf{x}}$; **(3)** $\infty$, if no adversary was found by $\mathfrak{a}$. Based on this definition we use four robustness evaluation metrics defined in [57, 60]:

**Median adversarial distance:** For each attack $\mathfrak{a}$ the median-$\delta_{\mathfrak{a}}$ score is defined as median $\{\delta_{\mathfrak{a}}(\mathbf{x}, y) \mid (\mathbf{x}, y) \in \mathcal{T}\}$, describing an averaged $\delta_{\mathfrak{a}}$ regarding $\mathcal{T}$ robust to outliers.

**Worst-case median adversarial distance:** The median-$\delta_p^*$ score is computed for all $L^p$-attacks as the median $\{\delta_p^*(\mathbf{x}, y) \mid (\mathbf{x}, y) \in \mathcal{T}\}$, where $\delta_p^*(\mathbf{x}, y)$ is defined as

Table 2: True positive and false positive rates for the four different rejection strategies. A threshold of $t_y = 0.4$ and $t_\beta = 0.3$ were used.

| database | | count | $t_y$ | $t_\beta$ | $t_y \vee t_\beta$ | $t_y \wedge t_\beta$ |
|---|---|---|---|---|---|---|
| clean images (FP) | correct classified | 9 938 | 2.0% | 3.8% | 4.1% | 1.7% |
| | incorrect classified | 62 | 58.1% | 91.9% | 91.9% | 58.1% |
| adversarial images (TP) | | 90 000 | 77.3% | 99.5% | 99.5% | 77.3% |

$\min \{\delta_\mathfrak{a}(\mathbf{x}, y) \,|\, \mathfrak{a} \text{ is a } L^p\text{-attack}\}$. This score is a worst-case evaluation of the median-$\delta_\mathfrak{a}$, assuming that each sample is disturbed by the respective worst-case attack $\mathfrak{a}_p^*$ (the attack with the smallest distance).

**Threshold accuracy:** The acc-$\mathfrak{a}$ of a model regarding $\mathcal{T}$ is defined as the percentage of adversarial examples found with $\delta_\mathfrak{a}(\mathbf{x}, y) > t_p$. This metric represents the remaining accuracy of the model when only adversaries below a given threshold are considered valid.

**Worst-case threshold accuracy:** The acc-$\mathfrak{a}_p^*$ of a model regarding $\mathcal{T}$ is defined as the percentage of adversarial examples found with $\delta_p^*(\mathbf{x}, y) > t_p$ using the respective worst-case attack $\mathfrak{a}_p^*$.

Similar to [57, 60] we used the following thresholds for the evaluation: $t_0 = 12$, $t_2 = 1.5$, and $t_\infty = 0.3$.

The robustness evaluation scores are presented in Tab. 1. As highlighted in bold, for all 9 attacks (including the worst-case) the CBC-4 with interpretable components outperforms all other models by a large margin. It must also be noted that the CBC-0 has by far the lowest robustness score across all attacks. This provides empirical evidence for the hypothesis that the interpretability of the components is a good indicator for the robustness of the CBC.

In general, the CNN baseline models have a higher robustness than the CBC-0. But in contrast to the relation between the CBC-0 and CBC-4, we observe no significant improvement in the robustness if we use the preferred feature extraction parameters in the CNN baselines. Hence, this suggests that the preferred standard setting is special to the CBCs.

**Rejection of adversarial examples** In Fig. 19 a collection of statistics is presented to highlight the difference between clean and adversarial examples for the CBC-4 model. Fig. 19a and Fig. 19c visualize the distribution of the predicted probability. The probability gap is visualized in Fig. 19b and Fig. 19d.

The distribution of adversarial images is similar to the distribution of incorrectly classified images. Hence, both have a significantly lower prediction probability *and* a smaller probability gap compared to correctly classified clean images. Using these two statistics, it should therefore be possible to construct a rejection strategy in which the false positives (rejected clean examples) are mostly incorrectly classified anyway. In Tab. 2 the True Positives (TP) and False Positives (FP) for four of such strategies are given. The *prediction reject strategy* rejects all inputs for which the predicted class hypothesis probability is below a probability threshold $t_y$. The *margin reject strategy* rejects all inputs with a probability gap below $t_\beta$. These two rejection strategies are combined using both the logical OR and AND operations.

From Tab. 2 we can conclude that it is indeed possible to use the class hypothesis probability and the probability gap of an image to efficiently reject adversaries. Using only the margin rejection strategy with $t_\beta = 0.3$ it is possible to reject close to all (99.5% of 90 000 samples) adversarial examples generated for the CBC-4 model with only a small cost in terms of FP rate (3.8%) of correctly classified clean samples. Only using the class hypothesis probability with $t_y = 0.4$ for rejection produces an even lower FP rate (2.0%) for correctly classified clean samples but fails to reject a large portion of the adversarial examples (22.7%). Due to the excellent performance when only the class hypothesis margin is used, not a lot of benefit in terms of FP and TP rates can be gained from combining the two strategies.[12]

(a) The distribution of the predicted class probability for correctly and incorrectly classified clean images.

(b) The distribution of the probability gap for correctly and incorrectly classified clean images.

(c) The distribution of the predicted class probability for clean and adversarial examples.

(d) The distribution of the probability gap for clean and adversarial examples.

Figure 19: Discrete approximations of the continuous density functions of different distributions of correctly / incorrectly classified and clean / adversarial images. Additionally, we highlight the used rejection thresholds.

A clear benefit from the combination of both rejection strategies can be observed when we apply the robustness evaluation performed in the previous section over the CBC-4 model in combination with the rejection strategies. Hence, an adversarial example is only considered as valid if it is misclassified and not rejected by the strategy. We realized the generation of such adversarial examples by updating the optimization goal of the adversarial attacks to the goal of fooling the reject strategy at the same time. The FOOLBOX implementations of the C&W, DEEPFOOL, and FGSM attack were consistently not able to find an adversarial example when the OR-combined rejection strategy was used. Non-gradient based approaches, which are less restricted by the inclusion of the reject strategy, required close to double the adversarial distance to fool both the model and the reject strategy compared to the previous robustness evaluation.

### E.2.5 Interpretation of the reasoning

The classification decision of a CBC can be easily interpreted. In the simplest form we can interpret the learned reasoning process by visualizing the reasoning probabilities and / or the learned components, as we showed multiple times before, e. g. in Sec. 4.1.1. If the CBC uses patch components, see Sec. 2.2, then it is also possible to visualize the learned model using heatmaps and reconstructions. We will focus in the following on the creation of these visualization and how they relate to the probabilistic model. While this section primarily focuses on the patch components setting, the visualization techniques can partly be applied to full-size component CBC, even when the components are not defined in the input space. By taking advantage of the similarity properties in use, we can construct further interpretation techniques, as was shown in Sec. E.2.2.

In this section, the same CBCs with patch components and spatial reasoning as in Sec. 4.1.2 are used. Again, they are denoted as $\alpha$-CBC and $\varnothing$-CBC. In general, the models follow the Siamese architecture depicted in Fig. 3 except that we apply a max pooling operation of pool size and stride $3 \times 3$ after the detection probability measuring to downsample the detection possibility stack to a size of $v_d, h_d = 7$. Both models share the same feature extractor of:

1. Convolution: 32 filters, kernel size $5 \times 5$, stride $1 \times 1$, bias, no padding;

2. Convolution: 64 filters, kernel size $3 \times 3$, stride $1 \times 1$, bias, no padding.

The 8 patch components are defined as $7 \times 7$ patches ($v_\kappa, h_\kappa = 7$) in the input space. Hence, the feature stacks have a size of $v'_x, h'_x = 22$ and $v'_\kappa, h'_\kappa = 1$ after the feature extraction. We applied multiple reasoning with two reasoning stacks per class and increased the number of training epochs to 300 accordingly. All other parameters, the training procedure and the initialization, are equivalent to the standard setting. The $\alpha$-CBC has trainable pixel probabilities $\alpha_{c,i,j}$ and the $\varnothing$-CBC has non-trainable, fixed pixel probabilities defined as $\alpha_{c,i,j} = 1/(v_r \cdot h_r)$.

**Theoretical basis of the visualizations** All visualizations show the learned reasoning process and, hence, the weights of the probability model. More precisely, we visualize the effective reasoning probabilities $\bar{\mathbf{r}}_c^{\pm}$ in two different ways. First, as input independent visualizations without the interaction with a given input. Second, as input dependent visualizations with interaction with a given input. CBCs compute the final probability output $p_c(\mathbf{x})$ for a class $c$ by Eq. (1) as the probability of agreement $A$ under the condition of importance $I$, see Sec. B.1, abbreviated by $A|I$:

$$P(A|I, \mathbf{x}, c) = (\mathbf{d}(\mathbf{x}))^\mathrm{T} \cdot \bar{\mathbf{r}}_c^+ + (\mathbf{1} - \mathbf{d}(\mathbf{x}))^\mathrm{T} \cdot \bar{\mathbf{r}}_c^-.$$

This equation can be split into positive agreement under the condition of importance $(A^+|I)$

$$P(A^+|I, \mathbf{x}, c) = (\mathbf{d}(\mathbf{x}))^\mathrm{T} \cdot \bar{\mathbf{r}}_c^+,$$

and negative agreement under the condition of importance $(A^-|I)$

$$P(A^-|I, \mathbf{x}, c) = (\mathbf{1} - \mathbf{d}(\mathbf{x}))^\mathrm{T} \cdot \bar{\mathbf{r}}_c^-.$$

Both entities resemble probabilities in favor of the class $c$. Particularly, $P(A^+|I, \mathbf{x}, c)$ is the probability in favor of class $c$ by components that have been detected and that should be detected (positive reasoning), and $P(A^-|I, \mathbf{x}, c)$ is the probability in favor of class $c$ by components that have been not detected and that should be not detected (negative reasoning). Additionally, we can compute the probability against a class $c$ by disagreement $\overline{A}$ under the condition of importance $I$, abbreviated by $\overline{A}|I$:

$$P(\overline{A}|I, \mathbf{x}, c) = 1 - P(A|I, \mathbf{x}, c),$$
$$= (\mathbf{1} - \mathbf{d}(\mathbf{x}))^\mathrm{T} \cdot \bar{\mathbf{r}}_c^+ + (\mathbf{d}(\mathbf{x}))^\mathrm{T} \cdot \bar{\mathbf{r}}_c^-.$$

This equation can be split into positive disagreement under the condition of importance $(\overline{A}^+|I)$

$$P\left(\overline{A}^+|I, \mathbf{x}, c\right) = (\mathbf{1} - \mathbf{d}(\mathbf{x}))^\mathrm{T} \cdot \bar{\mathbf{r}}_c^+,$$

and negative disagreement under the condition of importance $(\overline{A}^-|I)$

$$P\left(\overline{A}^-|I, \mathbf{x}, c\right) = (\mathbf{d}(\mathbf{x}))^\mathrm{T} \cdot \bar{\mathbf{r}}_c^-.$$

$P\left(\overline{A}^+|I, \mathbf{x}, c\right)$ can be interpreted as the probability against the class $c$ by components that have been not detected and that should be detected. Similar to that, is the interpretation of $P\left(\overline{A}^-|I, \mathbf{x}, c\right)$. It is the probability against class $c$ by components that have been detected and that should be not detected. These four probabilities are the main ingredients for the visualizations and are related to paths in the probability tree diagram $T$, see Fig. 2 for a sub-tree $T_c$ of $T$.

Figure 20: Detection probability heatmaps of the $\alpha$-CBC on a test sample. At the top of each heatmap we depict the respective patch component. We use the color coding "JET" to map probabilities of 0 to blue and 1 to red.

**Detection heatmap visualizations** We can use the detection probability function $d_k(\mathbf{x})$ for a certain component $\boldsymbol{\kappa}_k$ to get an indication where a component had a detection in the input. We visualize all eight detection probability heatmaps for the $\alpha$-CBC over a test input in Fig. 20. If we consider component b, then we see that it models almost a horizontal stroke. This is reflected in the heatmap showing that the horizontal lines of the two have a high detection to this component. Since the horizontal stroke is modeled in the upper part of the patch it might appear in the heatmap that the detection of the stroke is in a lower region and moreover has a slight offset. As another example, consider component g. This component is almost a vertical line and correctly detects the vertical part of the two. In contrast, the component d models the lower left part of a circle and, hence, has no similarities with parts within the sample.

The detection heatmaps are useful to gain an understanding of the similarity measure, especially if the representation of the components are learned in the input space and are interpretable. If this is not the case, then it could become hard to understand what a component really encapsulates. In this case applying a back projection strategy like in the IMAGENET experiment can be useful, see Sec. 4.2.

**Incorporation of pixel probabilities** If spatial reasoning is used, the pixel probabilities have to be incorporated in the visualizations. The pixel probabilities indicate how important a pixel position $i, j$ is for the class $c$. Moreover, they determine how to combine the single $p_{c,i,j}(\mathbf{x})$ for the class output $p_c(\mathbf{x})$. We include the pixel probabilities by a final scaling step applied to the reasoning heatmaps and reconstructions defined by:

1. Collect all the pixel probabilities $\alpha_{c,i,j}$ for one class $c$ into a map.
2. Normalize the map by $\max_{i,j} \alpha_{c,i,j}$.
3. Upsample the map to the respective size of the target map (e. g. a heatmap).
4. Multiply the resized map with the target map.

**Reasoning heatmap: input independent visualizations** To answer the question "What has a model learned about a certain class $c$?" we consider input independent heatmaps. The idea is to stimulate the effective reasoning possibility vectors $\bar{\mathbf{r}}_c^{\pm}$ by the optimal detection possibility vector and highlight the regions in favor of class $c$ from the positive and negative reasoning perspective. The optimal detection possibility vector for $A^+|I$ is $\mathbf{d}(\mathbf{x}) = \mathbf{1}$ and, moreover, $\mathbf{d}(\mathbf{x}) = \mathbf{0}$ for $A^-|I$. By using these optimal inputs we compute the reasoning heatmaps for a class $c$ by:

1. Upsample the $v_r \times h_r \times \#\mathcal{K}$ effective reasoning possibility stacks to $v_x \times h_x \times \#\mathcal{K}$.
2. Compute the pixel-wise dot product with the respective optimal detection possibility vector to obtain the heatmap of size $v_x \times h_x$.
3. Overlay the pixel probability map.

By these maps we find the regions for a model where components have to be and have to be not detected. Consider Fig. 21 where we visualize the learned concepts for both models. Fig. 21a shows the learned sparse encodings of the $\alpha$-CBC. For example for class 3, the model learns to classify the digit by recognizing specific line endings. In contrast, the class 1 is classified with negative reasoning only by checking that no component matches around the vertical stroke.

In Fig. 21b we depict the reasoning heatmaps for the non-trainable pixel probabilities model $\varnothing$-CBC. Since all $\alpha_{c,i,j}$ are equivalent, the final overlay of the pixel probability maps does not change the

(a) $\alpha$-CBC

(b) $\varnothing$-CBC

Figure 21: Input independent reasoning heatmaps for $\alpha$-CBC and $\varnothing$-CBC for all classes. For each class, we depict one reasoning stack. Class labels are shown at the top. We use the color coding "JET" to map probabilities of 0 to blue and 1 to red.

Figure 22: Input dependent reasoning heatmaps of $\alpha$-CBC and $\varnothing$-CBC for certain classes and for a test sample (class 7) from the MNIST database. We use the color coding "JET" to map probabilities of 0 to blue and 1 to red.

visualizations as each pixel probability has the same importance. Moreover, the model is forced to reason correctly over each position to reach a high output probability. By the probability model, the sum over positive and negative effective reasoning has to be one. Hence, $A^+|I$ and $A^-|I$ are complementary to each other. The reasoning heatmaps can be hard to interpret as they might highlight background with positive reasoning in case a background component was learned. If we consider the learned concept of the class 0, then we see how the model detects the outer shape of the zero with $A^+|I$ and the black middle with $A^-|I$. Another good example is the class 4. The shape of a four is clearly visible in the $A^+|I$ heatmap and via $A^-|I$ it looks at the top and bottom that nothing is detected there to avoid confusions to other digits like a nine.

**Reasoning heatmap: input dependent visualizations** The input dependent heatmaps are constructed in the same way as the input independent heatmaps except that we take as possibility vector the detection possibility vector from the respective position of the detection possibility stack $\mathbf{d}\left(\mathbf{x}\right)$.

| $c$ | 0 | 1 | 2 | 3 | 4 | 5 | 6 | 7 | 8 | 9 |
|---|---|---|---|---|---|---|---|---|---|---|
| $A^+\|I$ | | | | | | | | | | |
| $A^-\|I$ | | | | | | | | | | |

(a) $\alpha$-CBC

| $c$ | 0 | 1 | 2 | 3 | 4 | 5 | 6 | 7 | 8 | 9 |
|---|---|---|---|---|---|---|---|---|---|---|
| $A^+\|I$ | | | | | | | | | | |
| $A^-\|I$ | | | | | | | | | | |

(b) $\varnothing$-CBC

Figure 23: Input independent reasoning reconstructions for $\alpha$-CBC and $\varnothing$-CBC for all classes. For each class we depict one reasoning stack. Class labels are printed at the top.

Additionally, we highlight the input image in the background. Now, we can visualize $A^{\pm}|I$ and $\overline{A}^{\pm}|I$ resulting in four possible visualizations.

Consider Fig. 22 where we took a test sample from the MNIST database and show for five classes which parts provided evidence in favor of or against the class decision. In combination with the optimal heatmaps of Fig. 21 we can see where the decision of the respective class diverges from the optimum. For example, the $\alpha$-CBC requires for an input to be classified as a two, that it detects the top arc ending, the significant bottom left corner, and the bottom line ending. The failure of the model to detect these features is depicted by the $\overline{A}^+|I$ heatmap highlighting the areas where these features should be detected to support the decision of a two. Nevertheless, the model highlights in the $A^-|I$ heatmaps, that the negative reasoning requirements are fulfilled to be classified as a two. In contrast, if we consider the $A^+|I$ and $A^-|I$ heatmaps of class 7 we see that all requirements are fulfilled to be a seven, which is the correct output class. For this correct class almost no $\overline{A}|I$ is observed. Note for other classes, like the class 9, how the heatmaps correctly highlight the common features of the two classes.

The behavior of the $\varnothing$-CBC is similar to the $\alpha$-CBC except that it requires a correct reasoning over the whole image instead of reasoning over a few regions. First, note how the combination of the $A^{\pm}|I$ and $\overline{A}^{\pm}|I$ heatmaps for one class result in the respective input independent heatmap of Fig. 21. Consider the class 4 of the $\varnothing$-CBC. The $A^+|I$ heatmap highlights correctly that the vertical stroke of the seven could be the vertical stroke of a four too. Hence, the method reasons in favor of class 4 over this part of the digit. Nevertheless, it highlights in the $\overline{A}^+|I$ heatmap that the left part of the four is not detected. Additionally, the important sanity check that there is no top stroke at a four fails as it is clearly highlighted in the $\overline{A}^-|I$ heatmap. Overall the heatmaps of this class highlight a lot $\overline{A}|I$ for the classification decision of a four. In contrast, there is close to no $\overline{A}|I$ for the correct class and, hence, the input is correctly classified as a seven.

**Reasoning reconstruction: input independent visualizations** The reconstructions are similar to the heatmaps with the difference that we incorporate the learned patch components. A requirement for this visualization technique is that the components are defined in the input space. Moreover, we assume that the input space was defined over $[0, 1]$. Again we use the optimal detection possibility vectors and do the reconstruction by:

| $c$ | $\alpha$-CBC | | | | | $\varnothing$-CBC | | | | |
|---|---|---|---|---|---|---|---|---|---|---|
| | 0 | 3 | 5 | 6 | 8 | 0 | 3 | 5 | 6 | 8 |
| $A^+\|I$ | | | | | | | | | | |
| $\overline{A}^+\|I$ | | | | | | | | | | |
| $A^-\|I$ | | | | | | | | | | |
| $\overline{A}^-\|I$ | | | | | | | | | | |

Figure 24: Input dependent reasoning reconstructions of $\alpha$-CBC and $\varnothing$-CBC for certain classes and for the third test sample (class 0) from the MNIST database.

1. Initialize a zero matrix of size $(v_r + v_\kappa) \times (h_r + h_\kappa)$ as target image.[13]
2. For each pixel position $i$, $j$ in $v_r \times h_r$ and component $\kappa_k$ do:
   (a) Scale the component $\kappa_k$ by the probability $i$, $j$, $k$ from the effective reasoning possibility stack.
   (b) Add the scaled component of size $v_\kappa \times h_\kappa$ to the target image at the corresponding receptive field, which is the area $i, i+1, ..., i + v_\kappa - 1$ and $j, j+1, ..., j + h_\kappa - 1$.
3. Normalize each intensity value in the target image by the overall count, that a filter of size $v_\kappa \times h_\kappa$ has covered that pixel when it was slid over the target image.
4. Overlay the pixel probability map.

The visualizations which are obtained by this principle visualize the same information that are included in the corresponding heatmaps. Hence, it is just a different way of visualizing the learned concepts. The advantage is that the reconstructions for the $\varnothing$-CBC are easier to interpret. For example, Fig. 23b clearly shows digit shapes in $A^+|I$ for the most classes. Moreover, the learned sanity checks by $A^-|I$ becomes easier to understand. For the class 3 the model checks that the ends of the three are not closed, which is an important difference to an eight. In contrast to the heatmaps, the overlay of $A^+|I$ and $A^-|I$ does not result in a white image as we incorporated the components which could consist of black regions or not perfectly white shapes.

On the contrary, the $\alpha$-CBC reconstructions are pretty much black images with white blobs. It underlines the strong sparse coding which is learned for MNIST. For example, the classification of a six is based on the recognition of the intersection between the lower and the upper part, the top line ending, and that there is no line which connects the upper ending with the lower circle (to distinguish to an eight).

**Reasoning reconstruction: input dependent visualizations**  Similar to the heatmaps, we obtain these visualizations by exchanging the optimal possibility vector with the detection possibility vector from the respective position of the detection possibility stack $\mathbf{d}(\mathbf{x})$. Beside that we follow the algorithm for input independent reconstructions. In Fig. 24 we show the reconstructions of a test sample from the MNIST database. In accordance to that, we show the reconstructions for those classes which were not depicted in Fig. 22. Both models do a correct prediction as we see in the $\overline{A}^\pm|I$ reconstructions. Nevertheless, how they achieve this is highly different and we leave it to the interested reader to start an interpretation why the models did not classify the input as another digit.

**Summary**  In the previous paragraphs, we studied the interpretability of patch models with spatial reasoning using two different models. With an accuracy of $(97.33 \pm 0.19)\,\%$ the models perfor-

Figure 25: Results of the cut-off experiment where we modified a base image regarding the import regions marked by $A|I$ heatmaps.

mance is considerably lower than the state-of-the-art. One reason might be that the models we used are quite sparse. Both have only 35 K trainable parameters. The accuracy can be increased towards state-of-the-art if we use a deeper feature extractor. A problem that emerges in this case is that the receptive field and moreover the minimum size for patches increases too as the receptive field size is the minimum patch size. Of course, this is not a problem in general but it was obstructive in the experiments. If we increase the network depth and, moreover, the patch size, then the network starts to rely mostly on positive reasoning and, moreover, in some cases the reconstructions become less meaningful. However, the goal of this section was to show that the model classifies by using negative reasoning and that the reconstruction principle works. Hence, we decided to accept the low accuracy models.

The reconstructions of inputs via back projecting the components seems to be a visualization that should work in principle. However, till now we neither got it working to acceptable results on colored datasets nor on deep networks with more than 4-layers. The reason for that is not clear so far and we hope that we can improve these results in the future.

The heatmaps and the detection probability maps are applicable independently to the space where the components are trained and, hence, provide a powerful tool to gain insights of the learned reasoning process. To provide evidence that these visualizations are really showing the learned concepts, we present in the main part of the paper the visualizations with an adversarial image. The idea behind this experiment is to make a stress test of the evaluations by explaining "Why does the adversary fool the model?" In parallel to that, we can do an even more aggressive stress test: Because of Fig. 21 and Fig. 23 we know exactly which regions of a given digit are the most important parts for a certain classification. Hence, if the visualizations are correct, then it should be possible to remove all the unimportant parts without affecting the classification decision. Consider Fig. 25 where we visualize the results of such an experiment for both models. We used the same inputs for both models. The input sample is an image from the test sample database. Without any special tuning, we removed the parts of a three which are not marked as important by the $\alpha$-CBC see Fig. 21. As we see in the base $\mathbf{p}(\mathbf{x})$ distributions both models classify the original input image correctly but the $\alpha$-CBC predicts a more confident classification as the margins to other classes are higher. If we remove parts of the input which are highlighted to not contribute to the class decision of the $\alpha$-CBC for a three we observe no real drop in the output probability even though the probability for other classes changes. The image is still classified as a three even though it looks more like a small two. In contrast to the $\alpha$-CBC, the $\varnothing$-CBC realizes the manipulation as the probability drops but still classifies the image as a three. This is because in the $\varnothing$-CBC each pixel position contributes equally and, hence, every part of the three is important as shown in Fig. 23. This again underlines the descriptive power of the visualization techniques.

Figure 26: Visualizations of five learned components on GTSRB (left) and CIFAR-10 (right). *Top row:* The learned components. *Bottom row:* Similar training sample $\mathbf{x}$ for each $\boldsymbol{\kappa}_k$ regarding $d_k(\mathbf{x})$.

With the $\alpha$-CBC and $\varnothing$-CBC, we proposed two different models which behave totally different in classifying the data. The $\alpha$-CBC can classify the data by only reasoning over a few pixel positions. We consider this model type as an example of how usual NNs classify as we usually do not constrain a NN regarding how much of the image has to be understood. In contrast, the $\varnothing$-CBC is constrained to understand the whole image as each pixel position contributes equally. Right now, we are not sure what is the superior method (if there is one) or if a trade-off between both via a regularization of $\alpha_{c,i,j}$ is the right way to go. Remarkably, we have the feeling that the $\varnothing$-CBC could be the better method to detect outliers or manipulation in images, as in the experiments they always resulted in a drop of the probability $p_y(\mathbf{x})$ *and* the margin to the runner-up class (see also Fig. 5).

### E.3 GTSRB

In this experiment, we evaluate the performance and robustness of the CBC compared to a baseline CNN on GTSRB [61]. This is done to show that CBCs scale to RGB images with background noise where one class can be represented by a prototype. The 4-layer CNN feature extractor of the 43 components CBC is slightly different to the one used on MNIST:

1. Convolution: 32 filters, kernel size 7×7, stride 1×1, bias, no padding;
2. Convolution: 64 filters, kernel size 3×3, stride 1×1, bias, no padding;
3. Max pooling: pool size and stride 2×2;
4. Convolution: 64 filters, kernel size 5×5, stride 1×1, bias, no padding;
5. Convolution: 128 filters, kernel size 3×3, stride 1×1, bias, no padding.

The overall setting of the CBC is the standard setting for a CBC. We trained the network over increasing margins in three steps starting with $\beta = 0.1$ and increasing to $\beta = 0.2$ and $\beta = 0.3$. If trained from the beginning with the target margin of $0.3$, the model converges to a local minima of low accuracy. In this case, the model does not learn a strong positive component for each class. Moreover, some components resemble the same class. With only 43 components for 43 classes and the applied BMPP, this makes it impossible to correctly classify each class. We trained with each of the margins for 150 epochs. The images were rescaled to $v_x, h_x = 64$ and normalized to $[0,1]^{64 \times 64 \times 3}$ by the following procedure:

1. Normalize each image by the mean and standard deviation.
2. Clip the normalized image to the interval $[-2, 2]$.
3. Project the resulted image back to $[0,1]^{64 \times 64 \times 3}$.

The baseline CNN is constructed by replacing the detection probability and reasoning layer by a classic convolution and a fully-connected layer such that the CBC and the baseline CNN are architectural equivalent. To be more exact, the baseline CNN is obtained by applying the following two layers after the feature extraction:

1. Convolution: 43 filters, kernel size 22×22, stride 1×1, bias, no padding, ReLU;
2. Fully-connected: 43 neurons, softmax.

We trained the baseline CNN with the cross entropy loss for 450 epochs and with an initial learning rate of 0.0001. All other parameters are equivalent to the non-standard CBC setting.

Figure 27: Physical stop sign adversaries applied to the baseline CNN and to the CBC. The input **x** and the corresponding output distribution $\mathbf{p}\left(\mathbf{x}\right)$ are depicted. Additionally, the predicted label with the corresponding prediction probability are presented below each distribution. The first image is a sample from the GTSRB test database and shows the probability distribution of both models on a clean image.

Both networks achieve a comparable accuracy of $(97.2 \pm 0.77)\,\%$ for the CBC and $(97.5 \pm 0.18)\,\%$ for the baseline CNN. The trained CBC has a distribution of the probability gap similar to the 10 component CBC trained on MNIST. The average probability gap is $0.58 \pm 0.17$. In contrast, the baseline CNN has an average probability gap of $1 \pm 0.03$ and, hence, is almost returning a one-hot coding. As shown in Fig. 26 the trained CBC has interpretable components and classifies by the BMPP. In contrast to MNIST, we observed stronger variations in the components between the different runs. For example the background noise was highly varying. But in general, the components were always interpretable.

To underline the initial robustness evaluations performed on MNIST (see Sec. E.2.4), we tested the robustness of the CBC and the baseline CNN on the physical stop sign adversaries from [50]. The resulting behavior on these images is similar to the robustness results on MNIST. The CBC has no tendency to be overconfident, see Fig. 27. The CBC is not classifying all the adversaries correctly, but it does not predict an overconfident output probability either. Moreover, compared to the probability distribution on the clean input, we see a clear difference to the probability distributions of the adversaries. All probabilities drop and the model is clearly uncertain about its decision. This result is aligned to the observed behavior on MNIST and can be a promising property for further research in the field of outlier detection / rejection.

## E.4  CIFAR-10

Compared to GTSRB, CIFAR-10 [62] is harder to classify as the single classes cannot be represented by a single prototype due to large within-class variations and heavier background noise. We used a CBC with 10 full-size components and the standard setting, as found during the ablation study in Sec. E.2.2. To train on CIFAR-10, we changed the initialization strategy of the components. The 10 components are now initialized by using the 10 class means. Moreover, at the beginning we trained with the mean squared error over $\mathbf{p}(\mathbf{x})$ for 25 epochs against a one-hot class label vector. This helps the model to become discriminative. If trained over the mean squared error only, the network starts to converge to local minima of lower accuracies of $\approx 73\%$. After the initial few epochs, we switched to the training strategy used for GTSRB with increasing margins and an initial learning rate of 0.001. Again, the baseline CNN is the architectural equivalent of the CBC with the following layers after the feature extraction:

1. Convolution: 10 filters, kernel size 5×5, stride 1×1, bias, no padding, ReLU;
2. Fully-connected: 10 neurons, softmax.

The baseline CNN was trained with the cross entropy loss for 475 epochs and an initial learning rate of 0.001. All other parameters are equivalent to the non-standard CBC setting.

As expected, the CBC started to classify by the BMPP and formed prototypical components. We visualized five components in Fig. 26. The components show different common characterizing attributes of the classes, such as: texture (e.g. fur of the cat), rough shapes (e.g. main shape of the horse), important arrangements of features (e.g. ears, eyes, and snout of the dog), the general dominating colors at certain positions (e.g. white and blue for the ship), etc. Across all runs, the learned components looked almost equivalent.

The CBC achieves an accuracy of $(77.2 \pm 0.07)\,\%$ and the baseline CNN of $(79.9 \pm 0.3)\,\%$. Hence, the performance of both networks is almost at the same level. The probability distribution over the test dataset shows that the CBC has a much smaller average probability gap with $0.29 \pm 0.17$ than we observed for MNIST and GTSRB. This is also reflected in the low accuracy compared to state-of-the-art results. The fact that we had no success to "easily" train on CIFAR-10 in the same fashion in which we did on GTSRB, is a good indicator that the loss function might not be suitable in general and can be improved.