[Reviews · NeurIPS 2019]

Reviewer 1



Originality - this research is similar to prototype-based learning of neural networks, but it is the first to propose learning and detecting generic components that characterize object using three different types of reasoning (positive, negative and indefinite). Clarity - the paper is hard to read and follow. There are large chunks of text with no figures or equations to illustrate the concepts. In the supplementary material they provide a lot more information which was left out of the main paper. It does feel like the paper is not self-sufficient, as many important steps are only brushed over, such as the training procedure and how to generate the interpretations. Also, there are concepts such as "Siamese networks" which despite being known by most of the community deserve a citation or explanation for other readers. Same thing with the being more careful with definitions of variables (A and I are not defined). Also, the method used to generate the adversarial examples is not well explained. Quality: In terms of quality, my main issue was with the fact that the results do not seem to match the expectation set in the introduction that the model would be able to decompose objects into generic parts that operate as structural primitives (components). From the results it was not very clear to me what is the meaning of "component" and how to interpret the components that were extracted. In the case of digits it was easy to map each component to a digit and interpret the heatmaps as providing evidence for why an image would be classified or not as a digit. Even when the number of components is reduced to 9, we still interpret the components as digits but with one them missing, which forces the model to consider a digit as combination of other ones. However in the general case of images, the examples of visualization (figure 5) show "components" that look very similar to the training examples (or like "prototypes") and not like generic parts. The authors do not comment on this, which left me confused. Significance - a neural network architecture that can be interpreted through probabilistic reasoning (at least for the final layers) is likely to be built upon for further research, given the current interest in interpretable models.

Reviewer 2



Comment following the rebuttal: Dear authors, thank you very much for your response! I've read it and I'm happy to see the initial additional results on ImageNet like the component visualization at the top I think these are really useful. Overall after reading the paper, all the other reviews and the rebuttal, I feel I couldn't raise my score mainly because of the valid concerns raised by the other reviewers. The paper proposes a new type of recognition model: Given an image, it classifies it by performing a probabilistic reasoning process that considers the existence or absence of various components - indicative aspects of the object, and then combines together the evidence in a probabilistic (and thus potentially more interpretable) manner to output a final classification. The idea is interesting, and multiple experiments are provided to support it. The paper itself is also well-written and clear. - The related work section mentions prior work about Siamese architectures and Prototype classification, and also mentions that the idea about patches has been explored in past work. It would be great to add at least a brief discussion of how the new model proposed here differs from these prior approaches. - It would be nice to run ablations/variants of the model having e.g. only positive+neutral or only positive+negative, or even negative+neutral, and observe how each of these settings affects both quantitative performance and qualitative behavior. - Another potential direction could be performing any sort of a head-to-head comparison to a standard CNN. It may be the case that standard filters will also in practice learn to recognize the existence / absence of different patterns. Any sort of head-to-head comparison, either of how the filters vs components look, or measuring any numerical properties of them, could provide an important insight into how the new model compares to classic CNNs. - Continuing on this point, in the related work section, the first paragraph explains how standard neural networks can be interpreted as performing similar sort of reasoning where the weight signs indicate whether the reasoning is positive/negative. Then, it says that the use of ReLU is what prevents such an analogy. But using ReLU instead of e.g. sigmoid/tanh is exactly what made CNNs training better and faster and increased their stability [1]. Even though it doesn't have the probabilistic interpretation as in your model, it proved to work extremely well. Going “a step backwards” in a sense to constrained squashed activations makes me wonder about the scalability of this approach. Indeed there are experiments over ImageNet that explore a variant of the model, which is great, but I think further experiments that compare in this case e.g. learning curves, convergence rates, or again any other additional comparisons with CNNs would be very useful. - Looking at the examples in figures 3 and 5 makes me wonder to what degree the model may in practice find good clustering of the data and capture each center as a template/component, in which case the model might perform in a sense a bit of a nearest-neighbor search to one of the templates and classify the image to the closest center (?) Then, such an approach will work well when there are few object classes (MNIST, CIFAR10), but I think it is particularly interesting to explore further the behavior of the model when there is a much larger number of object types to recognize. Again, the paper indeed provides some experiments for ImageNet mentioning it gets comparable performance to existing approaches, but here also I think that actually more qualitative experiments over ImageNet would be really useful to understand how the model scales. - For the classification part, why did you choose contrastive loss and not e.g. log-likelihood loss? It would be good to either provide an intuitive explanation for that, or experiments with both types of losses. [1] Krizhevsky, Alex, Ilya Sutskever, and Geoffrey E. Hinton. "Imagenet classification with deep convolutional neural networks." In Advances in neural information processing systems, pp. 1097-1105. 2012.

Reviewer 3



This paper is well written, and I especially liked the introduction section and conclusions. But, not convinced by the proposed approach and its experimental study. It seems to me that the proposed approach makes a neural network model even more complicated. Although, one may define variables for particular purpose in an introduced model, it doesn't ensure that is what the variables accomplish when employed in complicated settings. The prcedure of extracting features, and detecting a component doesn't seem simple to me, and more like a black box. What is the motivation for using Siamese network for detecting a component? The paragraph, "Extracting the decomposition plan", could benefit from an expansion, possibly supplemented by a figure explaining the architecture that is explained in the paragraph. As for the types of reasonings are concerned, it doesn't have to be of the three kinds mentioned in the paper. Models like decision trees have more sophisticated reasonsings, yet interpretable. (A relevant paper: Simultaneous Learning of Trees and Representations for Extreme Classification and Density Estimation, ICML 2017). Most of the experiments section is devoted to MNIST dataset. This is a dataset for which components detection is relatively simpler, corresponding to even a single pixel. Even simple supervised models, and unsupervised models, work perform fine on these datasets. As for interpretability is considered, I don't see how the proposed approach is relevant for datsets like CIFAR-10, GTSRB. I don't think Fig. 4 is demonstrating the value of the model, in the sense of performing reasoning over components. Why not have large number of components? The number of components is chosen to be close to the number of classes? This doesn't seem intuitive. What is the explanation behind it? In fact, there is a hierarchy of components required for many computer vision tasks, as accomplished by CNNs. The proposed models seems a like a wrapper around existing neural approaches, for the namesake of interpretability. Should there not be a comparison w.r.t. other models claimed to be interpretable? There are many techniques just for interpreting a so called black box neural networks, for instance attention based techniques. Section 4.1.2 is too lengthy and complex to understand; the model should be such that it leads to interpretability; as per the section, there is a reliance of older techniques to understand the functioning of the model. It would have been more interesting if the proposed model is learned on a relatively less studied datasets, and identifies components interpretable to experts in that domain. The datasets considered in the paper, especially MNIST, have been studied in thousands of papers, with to many architectures explored for the datasets. So it becomes difficult to judge interpretability capabilities of a model, based upon evaluation of such a dataset, as it is highly subjective. Of course, it is good to use the dataset for a preliminary validation. ------------------------------------------------- I have taken the rebuttal into account as well as other reviews. I am fine with acceptance of the paper, assuming the authors would revised the draft as promised in the rebuttal draft.

[Author Response · NeurIPS 2019]

| Positive | | | | | | Negative | | | | | Indefinite | | | | |
|---|---|---|---|---|---|---|---|---|---|---|---|---|---|---|---|
| 1.0 | 1.0 | 0.81 | 0.78 | 0.77 | | 1.0 | 1.0 | 1.0 | 1.0 | 1.0 | | 0.32 | 0.31 | 0.31 | 0.31 | 0.30 |

Figure 1: The 5 components with the highest reasoning probability for each type of reasoning for 'Newfoundland dog'.

*Thanks to all reviewers for the effort and the patience leading to such interesting questions and very helpful comments to improve the quality of the contribution. As most reviewers highlighted similar questions and areas within the paper, we have chosen to discuss the comments in three main themes. But first we will give some general clarification.*

**Clarification:** The essential new idea proposed in CBCs is the probabilistic constraint on both, the penultimate and the final layer of feedforward NNs for classification, as stated in the conclusion. Both support the interpretability of the network. The main complexity of CBCs originates in the feature extractor which is used to improve the component detection. In general, the CBC is not restricted to a CNN as feature extractor. The use of a Siamese architecture is also not a hard requirement. As it can be seen in the experiments on IMAGENET presented here and in the paper, a similar interpretability of the components can be achieved without a Siamese architecture. Please excuse that we have an incorrect translation in the current version of the supplementary that we will correct: *We define our probability space over a set of events related to a probability tree diagram and not to a decision tree.*

**1) Clarity of the paper:** To improve the clarity and self-containment of our paper, we propose to use the additional page of the final version to move the class-dependent probability tree diagram (Fig. 6 in the supplementary material), that models the class hypothesis probability $p_c(\mathbf{x})$, from the supplementary to the main part of the paper and provide a short mathematical derivation and a short explanation of the training procedure accordingly. This will provide clarity regarding the random variables (e.g. $A|I$) and improve the explanation of the visualizations in Sec. 4.1.2., as these can then be directly related to paths in the tree diagram. Including the probability tree diagram also clarifies how BIEDERMAN's RBC-theory is realized and the distinction between CBCs and the modeling of knowledge in general graph structures, e.g. in the mentioned ICML paper about the learning of trees. Opposed to general modeling of knowledge in graphs, CBCs only rely on the proposed kinds of reasoning and assume that components are stochastically independent. These assumptions are needed to keep the method simple. Moreover, we agree that our related work section has to be extended and that additional cross-references and citations have to be included to improve the paper.

**2) Extended evaluation of complex datasets:** We acknowledge that the experimental results presented in the paper are lacking complexity. This is partly due to the introductory nature of the paper. The datasets were chosen such that they reflect the functionality of different parts of the proposed approach. The experiments on MNIST show that the CBC architecture can fulfill the goals in principle. CIFAR-10 and GTSRB show that the model is capable of learning color components and IMAGENET shows the scalability of the approach. We have however neglected to show parts of the approach working on more complex datasets. To solve this, we plan to extend the evaluation on IMAGENET. To make space for this, we will move the experiments on CIFAR-10/GTSRB to the supplementary. In Fig. 1 the proposed extension of the experiments is shown. Using the discussed back projection method, we have computed a respective mapping of the $2 \times 2 \times 2048$ components used in the detection probability function to components that are part of the dataset. The proposed extension shows the five components with the highest reasoning probabilities for positive, negative, and indefinite reasoning for the class 'Newfoundland dog'. Investigating these components leads to a deeper understanding of the model's classification. For example the split into different types of dog snouts over positive, negative and indefinite components shows the importance of the type of dog snout for the classification. Notably, special breeds of the Newfoundland dog are known for their white marks. This characteristic is modeled by the first positive reasoning component. Considering that not all breeds of the Newfoundland dog have these marks, this might indicate a bias in the IMAGENET dataset.

We believe that this extension of the experiment on IMAGENET delivers further insight into the interpretability that the method provides, even without the usage of a Siamese architecture. Moreover, note that Fig. 1 is an example illustration. The figure for the final paper will also contain an input image of the class and make a comparison to components of other similar classes. If space allows, we will also move the learned MNIST patch components in Fig. 20 to the main part of the paper. These images highlight the decomposition of an image by the CBC.

**3) Comparison to other methods:** Thank you for the provided suggestion regarding additional comparisons. A first initial comparison to CNNs has already been presented in the supplementary during the ablation study. Moreover, we investigated a comparison of the CBC agreement heatmap visualizations to CAM methods. Disagreement cannot be compared to CAM methods as they cannot provide a similar visualization. However, such comparisons will be part of future contributions as well as a detailed study about adversarial robustness and outlier detection.

[Meta-Review · NeurIPS 2019]

The paper proposes an interesting probabilistic reasoning process that considers the presence or absence of various components (that are indicative of several properties of an instance) and combines them together as (potentially interpretable) evidence for its final classification. The idea seems to us novel and interesting. Multiple experiments are provided to support the approach. The paper is also well-written and clear. Hence, we recommend acceptance.